# Induced neural phase precession through exogenous electric fields

Miles Wischnewski [1,4] ✉, Harry Tran[1,4], Zhihe Zhao [1], Sina Shirinpour[1], Zachary J. Haigh [1], Jonna Rotteveel[1], Nipun D. Perera [1], Ivan Alekseichuk [1], Jan Zimmermann [1,2,3] & Alexander Opitz [1] ✉

The gradual shifting of preferred neural spiking relative to local field potentials (LFPs), known as phase precession, plays a prominent role in neural coding. Correlations between the phase precession and behavior have been observed throughout various brain regions. As such, phase precession is suggested to be a global neural mechanism that promotes local neuroplasticity. However, causal evidence and neuroplastic mechanisms of phase precession are lacking so far. Here we show a causal link between LFP dynamics and phase precession. In three experiments, we modulated LFPs in humans, a non-human primate, and computational models using alternating current stimulation. We show that continuous stimulation of motor cortex oscillations in humans lead to a gradual phase shift of maximal corticospinal excitability by ~90°. Further, exogenous alternating current stimulation induced phase precession in a subset of entrained neurons (~30%) in the non-human primate. Multiscale modeling of realistic neural circuits suggests that alternating current stimulation-induced phase precession is driven by NMDA-mediated synaptic plasticity. Altogether, the three experiments provide mechanistic and causal evidence for phase precession as a global neocortical process. Alternating current-induced phase precession and consequently synaptic plasticity is crucial for the development of novel therapeutic neuromodulation methods.

The timing of neural spiking is often coupled to ongoing oscillating local field potentials (LFPs)[1,2]. LFPs represent the spatiotemporal sum of extracellular electric fields in the brain[3]. Oscillatory extracellular electric fields can reciprocally determine spike timing by mechanisms such as ephaptic coupling[4,5]. This coupling between neural firing and extracellular fluctuations is referred to as entrainment[6]. That is, spiking predominantly occurs at a specific phase of the extracellular oscillation. Notably, this phase preference can change over time[7–13]. Such a shift in spike timing relative to LFPs is hypothesized to transfer information about behavioral processes[8,14,15]. Known as phase precession, gradual shifts in the timing of neural firing relative to the LFP were initially found in hippocampal place cells and entorhinal grid cells[7,9].

For example, a gradual shift in spike preference observed in rats during spatial navigation reflects the encoding of different locations within a maze[7,9,16]. Recently, phase precession was demonstrated in humans using invasive recordings[8,17]. Further, phase precession has been shown to manifest in a wide range of cortical and subcortical regions[18–23], indicating a global neural mechanism that promotes synaptic plasticity and ultimately enables learning[14,15,24–30]. However, to date, evidence for this hypothesis is mostly correlational. That is, there has been a lack of causal evidence, such as the direct modulation of phase preference in neural firing.

Given the codependence between spike timing and LFPs it could be expected that exogenous modulation of LFPs influences phase

[1]Department of Biomedical Engineering, University of Minnesota, Minneapolis, MN, USA. [2]Department of Neuroscience, University of Minnesota, Minneapolis, MN, USA. [3]Department of Radiology, Center for Magnetic Resonance Research, University of Minnesota, Minneapolis, MN, USA. [4]These authors contributed equally: Miles Wischnewski, Harry Tran. ✉e-mail: mwischne@umn.edu; aopitz@umn.edu

precession and related brain functions. An emerging scientific method to achieve such modulation is via exogenously applied low-intensity alternating currents (AC). A plethora of studies in cellular slices[5,31–34], rodents[35–38], and non-human primates (NHP)[39–42] show that the application of AC at frequencies corresponding to the neurons' natural firing rhythm can entrain neurons and change spike-timing[43]. Further, the non-invasive application of AC through electrodes on the scalp can result in physiological and behavioral changes in humans[44–48]. For example, the use of transcranial alternating current stimulation (tACS) was found to improve working and long-term memory[49–53], as well as learning[54–58]. Further, tACS effects can outlast the stimulation period and have been linked to synaptic plasticity[59–61].

Outside modulation is thought to cause the LFP to adapt to the exogenous field[35,40,41]. Thus, preferred spiking at a specific endogenous oscillation phase would transition to spiking at the corresponding exogenous oscillation phase[39]. A question that follows is whether the modulation of LFPs through AC stimulation can prompt a phase shift in neural output. Evidence towards this notion would advance our understanding of phase precession in humans and how it relates to functional processes. For this, three key questions must be addressed: First, are phase preferences of neural firing shifted during application of external AC? Second, are these changes in spiking phase preference consequential for the system-level brain functions? Third, can observed phase shifts be explained through neuroplasticity mechanisms? To address these questions, we combine evidence from human experiments, invasive recordings in a NHP and neuronal network modeling. We investigated the effects of AC neuromodulation at both macroscopic and microscopic scale. Additionally, we developed a computational model to translate from single unit physiology to network activation. We found that overall cortical output and a subset of neurons are phase-entrained during tACS. Furthermore, the preferred phase for cortical output was gradually shifted during tACS. Together, we provide comparative evidence in humans, in NHPs, and in silico for induced phase precession of single neurons and neuronal populations in the neocortex by electric fields. The recognition of phase precession as a global brain process connected to neuroplasticity, which can be externally altered, could prove crucial for understanding the effects of neuromodulation therapies in neurological and psychiatric disorders.

## Results

### Experiment 1: AC stimulation modulates cortical excitability in humans

Alternating currents were applied in healthy human volunteers to modulate direct motor cortical output to the right-hand muscle (Fig. 1a). Oscillatory electric fields were applied through two electrodes placed on the scalp anterior and posterior of the motor cortex at frequencies mimicking the endogenous sensorimotor alpha (7–13 Hz) and beta (14–30 Hz) rhythm. The induced electric field strength in the precentral gyrus was 0.31 mV/mm based on electric field modeling (Supplementary Fig. S3). Polarization of pyramidal neurons in the anterior and posterior precentral gyrus depends on the AC phase. During the AC phase in which the electric current direction is posterior-anterior (PA; from here on referred to as 0°) the soma of pyramidal neurons in the primary motor cortex, i.e., Brodmann areas (BA) 4a and 4p are depolarized. During the AC phase where electric current direction is anterior-posterior (AP; from here on referred to as 180°) soma in the premotor cortex, i.e., BA6 are depolarized (Fig. 1d, e).

We found that motor cortex excitability significantly depends on AC phase ($F = 8.62$, $p = 8.30e-5$, $\eta_p^2 = 0.312$; Fig. 2a, b). This effect did not differ between the two frequencies ($p > 0.3$). At 90° and 180° of the AC phase cortical excitability was significantly increased ($p < 0.04$, Cohen's $d > 0.5$), whereas at 0° excitability was decreased ($p = 0.001$, Cohen's $d = 0.86$). These data show that rhythmic depolarization of the premotor cortex at alpha and beta frequency yields an increase in descending volleys towards the muscle. This suggests a transsynaptic

effect of AC that is mediated by premotor cortex, whereas direct entrainment of monosynaptic primary motor cortex neurons leads to decreased motor output. We explored this possibility further using computational modeling, as is discussed below.

Further, we observed that the optimal phase for motor output shifts over time. We calculated the polar vector strength and direction of the average normalized cortical excitability over AC phases. Note that larger values of polar vector strength (non-uniformity) relate to a stronger bias towards a specific phase. These estimates were obtained using a moving average (sliding window length: 55 trials, ~120 s, 20 steps of 5 trials, averaged over four blocks). For alpha and beta stimulation we observed a gradual forward moving phase shift (circular-linear correlation, alpha: $r = 0.655$, $p = 0.014$, beta: $r = 0.825$, $p = 0.001$). The preferred phase for motor excitability starts at 92.7° (window 1) and moves to 163.6° (window 20) when alpha stimulation is applied (Fig. 2c). Preferred phase for motor excitability starts at 131.8° (window 1) and moves to 189.2° (window 20) during beta stimulation (Fig. 2d). Permutation testing on excitability with randomized phases suggested that spurious uniform phase shifts in our data are unlikely ($p < 0.05$, Supplementary Fig. S5). Also, we investigated potential phase shifts to a virtual tACS signal in a dataset where no active stimulation was applied, and found no apparent phase preference, nor phase shifts (Supplementary Fig. S6). Furthermore, phase shifts were consistently observed within each stimulation block and a reset of phase was observed between blocks (Supplementary Fig. S7). Resampling with N-2 subgroups suggested that the observed phase shifts were not driven by outliers (Supplementary Fig. S8).

Besides a shift in preferred phase, we also observed a general increase in excitability during stimulation blocks (linear Pearson correlation, alpha: $r = 0.722$, $p < 0.001$; beta: $r = 0.552$, $p < 0.001$; Supplementary Figs. S9, S10). This effect was absent when no stimulation was applied ($p > 0.5$). These increases in excitability were independent of phase shifts (Supplementary Fig. S11). Furthermore, the observed changes in phase preference and excitability are not associated with subjective measures on participants' arousal (Supplementary Fig. S12). The observed phase shift in cortical excitability is reminiscent of single-cell phase precession[7,9]. Whether AC stimulation can induce phase precession in single units was investigated in an awake non-human primate (experiment 2).

### Experiment 2: AC stimulation modulates neuron spiking activity in the non-human primate brain

We recorded single unit activity using a 128-channel microdrive recording system implanted over the left hemisphere in an awake non-human primate (Supplementary Fig. S13). AC stimulation (10 and 20 Hz) was then applied through two scalp electrodes positioned on the frontal and parieto-occipital area (Fig. 1b). The electric field strength had a maximum value of 0.84 mV/mm and was between ~0.1 and ~0.75 mV/mm in the region that contained the recording electrodes (Supplementary Fig. S14). Using an offline spike sorting method, we identified the spiking activity of 81 single units (Supplementary Fig. S15).

Out of a total of 81 neurons, 46 (56.8%) and 48 (59.3%) were significantly entrained during AC stimulation at alpha and beta frequencies respectively (Fig. 3a, g), as shown by Rayleigh's test for non-uniformity (alpha: $t(45) = -5.48$, $p = 1.85e-6$; beta: $t(47) = -5.57$, $p = 1.20e-6$). The phase locking value (PLV) of responsive neurons increased during stimulation compared to the pre-stimulation period for both frequencies (alpha during: $0.20 \pm 0.028$ vs alpha pre: $0.05 \pm 0.004$, Fig. 3d; beta during: $0.22 \pm 0.20$ vs beta pre: $0.06 \pm 0.04$, Fig. 3j). The firing rate was not significantly different between the pre-stimulation, during stimulation and post-stimulation for both alpha and beta tACS ($p > 0.15$, Fig. 3c, l).

To classify neuron behaviors based on their phase shifts, we created a framework that could be divided into three steps

(Supplementary Fig. S16): (1) we keep neurons that are responsive in at least 10 time windows (or 50% of the total number of time windows), (2) we select neurons that exhibit a significant circular-linear correlation and finally (3) we keep neurons exhibiting a phase shift greater than |15°| between the preferred phase in the first time window and the last time window. We found that several entrained neurons exhibit a shift in preferred phase over time during the stimulation block. This was observed for both AC frequencies. As in experiment 1, we estimated phase shifts using a moving average (sliding window length: 132 s, 20 windows, 12 s step size, averaged over four blocks). Only neurons exhibiting a phase shift of >15° and a circular-linear correlation greater than 0.5 were included in the analysis (alpha stimulation:

$n = 15$, mean $r = 0.80 \pm 0.031$; beta stimulation: $n = 13$, mean $r = 0.81 \pm 0.081$, but see Supplementary Table 2). For alpha stimulation, 8 neurons showed a clockwise (negative) phase shift (mean: −38.26°), meaning that neural spiking was pushed backward in the oscillatory cycle by AC stimulation (Fig. 3b, e, Supplementary Figs. S17A, S19). We observed a counter-clockwise (positive) phase shift in 7 neurons (mean: 31.94°) suggesting that neural spiking preference moved to later in the oscillatory cycle (Fig. 3b, f, Supplementary Figs. S17B, S19). The maximal and minimal value is respectively 76.92° and −57.75°. The remaining 31 entrained neurons did not show a uniform phase shift. This means that they either showed a stable phase preference (Supplementary Fig. S20A) or showed phase shifts in more than one

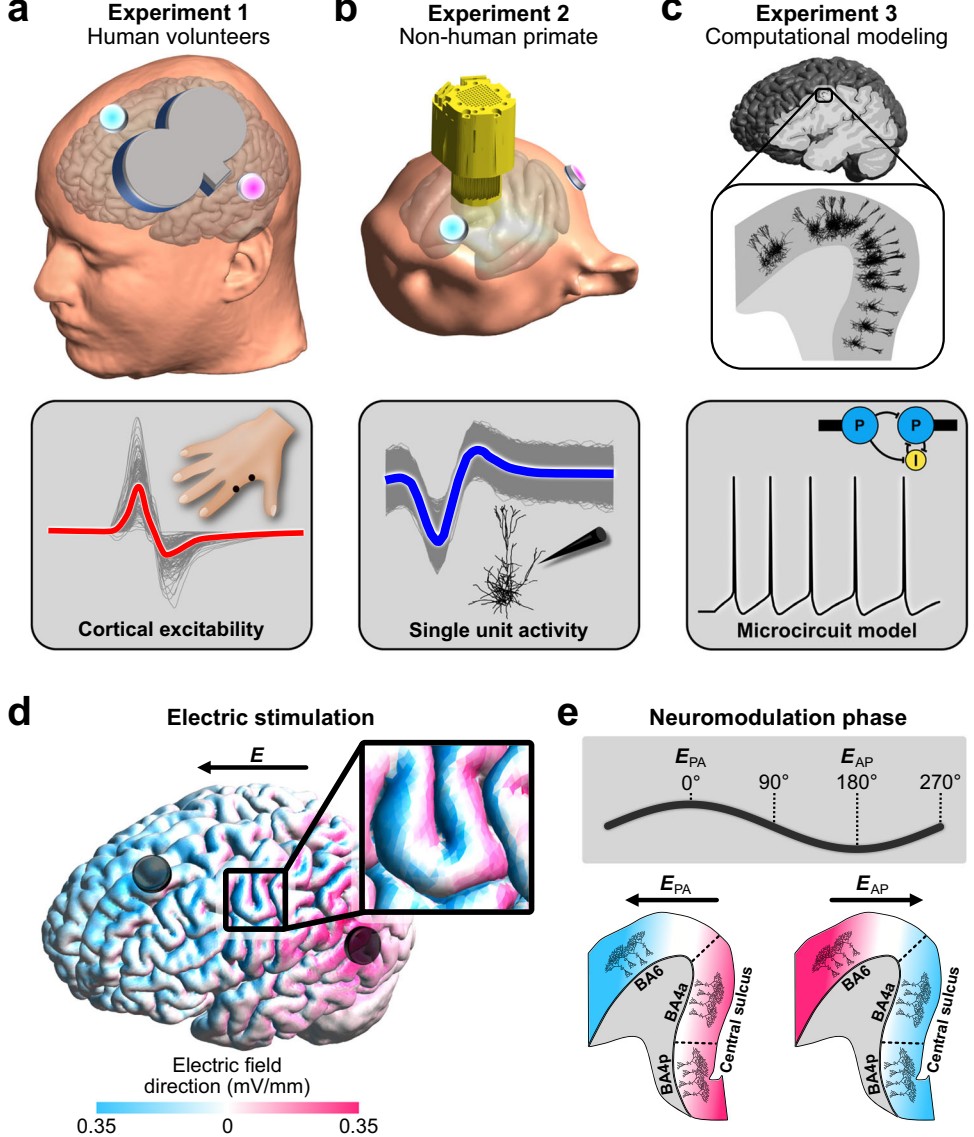

**Fig. 1 | Overview of the three experiments. a** In experiment 1, two sessions of AC stimulation targeting the motor cortex (intensity: 2 mA peak-to-peak, frequencies: 9.92 ± 0.25 Hz and 20.24 ± 0.89 Hz; Supplementary Fig. S4) were performed in 20 healthy human volunteers. Cortical excitability was probed using single-pulse transcranial magnetic stimulation, which resulted in a motor-evoked potential in the first dorsal interosseous muscle. **b** In experiment 2, AC stimulation (intensity: 2 mA peak-to-peak, frequencies: 10 and 20 Hz) was performed in a non-human primate implanted with 128 microelectrodes to record neural spiking (left frontal cortex covering motor to prefrontal areas). **c** In experiment 3, we used multi-scale computational modeling to investigate the effect of AC stimulation on spiking

activity and neuroplastic changes. **d** Electric field direction with respect to the gyral surface at the 0° tACS phase. Inward current flow is shown in magenta and outward current flow is shown in blue. **e** The phases of AC stimulation relate to differences in current direction. At 0° current direction is posterior-to-anterior (PA) and at 180° current direction is in anterior-to-posterior (AP). In the lower panel a sagittal depiction of the precentral gyrus is shown at different AC phases. AC stimulation at 0° primarily depolarizes soma in Brodmann area 4a and 4p (primary motor cortex). AC stimulation at 180° primarily depolarizes soma in Brodmann area 6 (premotor cortex).

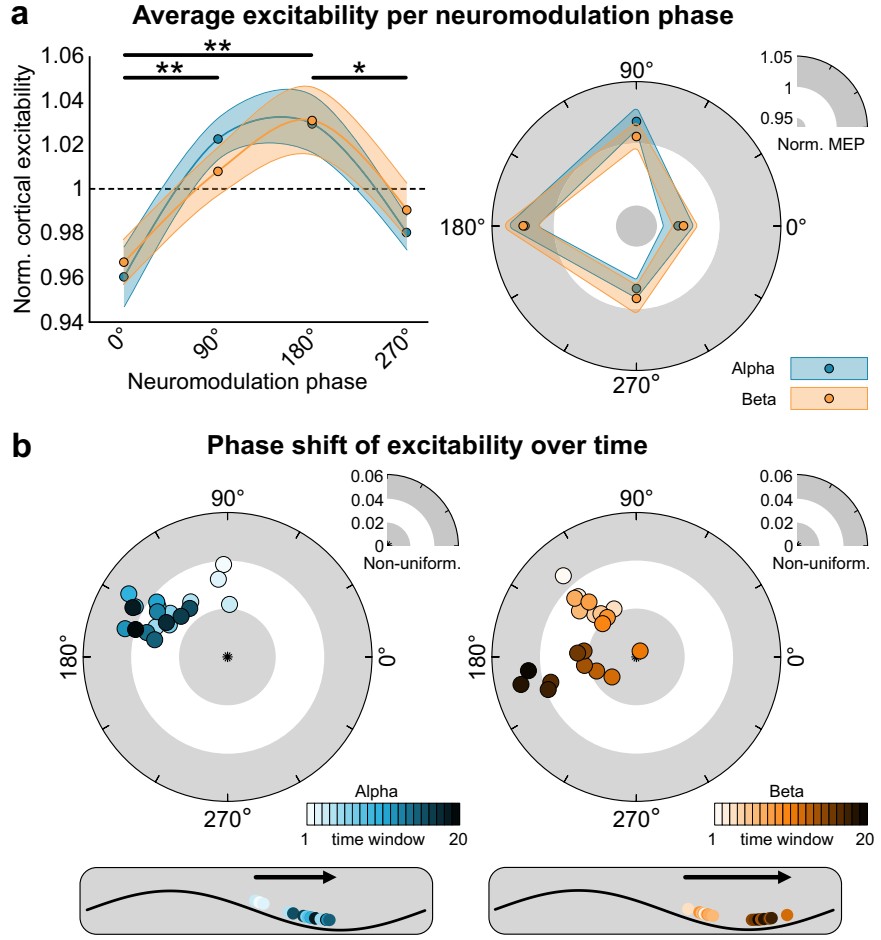

**Fig. 2 | Effects of AC stimulation on cortical excitability in humans. a** Average excitability per neuromodulation phase in a Cartesian (left) and polar (right) plot ($n = 20$ human participants). Motor cortex excitability was probed at four phases of the tACS oscillation (0°, 90°, 180°, and 270°). Motor cortex excitability is defined as the normalized motor-evoked potential following a single transcranial magnetic stimulation pulse. Averaged over a total of 600 trials (~150 per phase), there is a significant effect of phase (2 × 4 repeated measures ANOVA; $F = 8.62$, $p = 8.3e\text{-}5$), with higher excitability at 90° and 180° compared to 0° (paired t-test (two-sided, Bonferroni-corrected); $t = 3.60$, $p = 0.004$; $t = 4.63$, $p = 1.8e\text{-}4$ respectively), as well as for 180° compared to 270° (paired t-test (two-sided, Bonferroni-corrected); $t = 3.11$, $p = 0.017$). The phase-dependency was similar for AC applied at the alpha (blue) and beta (orange) frequency. Shaded areas reflect standard error of mean.

$*p < 0.05$, $**p < 0.005$. **b** Polar plots of changes in phase of primary motor cortex excitability over time averaged over participants ($n = 20$ human participants). A sliding time window was used on the polar vector of the motor-evoked potential amplitudes collected at the four phases, averaged over four blocks. Each dot represents the polar vector of 55 trials, which equates to approximately two minutes. The sliding window moves in steps of 5 trials (~12 s), resulting in 20 windows. Results show phase shifts both alpha and beta stimulation over time (circular-linear correlation, alpha: $r = 0.655$, $p = 0.014$, beta: $r = 0.825$, $p = 0.001$). Preferred phase for primary motor cortex output starts at 92.7° (window 1) and moves to 163.6° (window 20) when alpha stimulation is applied. During beta stimulation starts phase preference shifts from 131.8° (window 1) to 189.2° (window 20). Source data are provided as a Source Data file.

direction over time (Supplementary Fig. S20B). For beta stimulation, 4 neurons showed a clockwise (negative) phase shift (mean: −38.51°; Fig. 3h, k, Supplementary Figs. S18A, S20) while 9 neurons showed a counter-clockwise (positive) phase shift (mean: 27.23°; Fig. 3b, e, Supplementary Figs. S18B, S19). The maximum and minimum value is respectively 69.13° and −67.16°. The remaining 35 responsive neurons showed no or non-uniform phase shifts (Supplementary Fig. S21). Of the 28 neurons that showed significant phase precession during AC stimulation (alpha $n = 15$, beta $n = 13$), none were found to display significant phase shifts in a 6-min period without stimulation (Supplementary Fig. S22). Analysis of phase shifts across all entrained neurons confirms this observation (Supplementary Fig. 23). Furthermore, phase shifts were consistent across the four blocks and phase was reset between blocks (Supplementary Fig. S24), comparable to what was observed in the human data (Supplementary Fig. S7). Results for different angle thresholds are shown in Supplementary Table 2. Together these results suggest that a subset of neurons displays phase precession-like behavior in response to an external AC field[7,9].

## Experiment 3a: spatial distribution of entrainment

In our multiscale computational modeling approach (experiment 3a), we first simulated entrainment effects by populating the head model with realistic model neurons in a region of interest consisting of the left motor hand knob, including BA4 and BA6 (Supplementary Figs. S25, S26). For this simulation the tACS electrodes were placed concordant to experiment 1. Within the modeled region, the electric field is strongest at crown of the precentral gyrus and decreases with depth into the sulcus (Supplementary Fig. S3). To investigate the rhythmic membrane de- and hyperpolarization effects on spiking activities, we determined the effect of tACS on the firing rate during and without stimulation. Synaptic inputs were tuned for cells to have similar intrinsic firing rates around 10 and 20 spikes per second, based on the assumption that these values contribute to the endogenous alpha and beta oscillations and are more susceptible to stimulation at these frequencies[38,62,63]. The overall firing rates of all model neurons in the pre-stimulation baseline period and during the stimulation were numerically equivalent (Supplementary Fig. S27).

To quantify the tACS effect on neural entrainment, we calculated the PLV for all cells. Layer 5 (L5) pyramidal neurons showed significantly increased PLV during AC stimulation compared to the pre-stimulation baseline period (c), suggesting increased entrainment

(Fig. 4c). Crucially, neural entrainment during AC stimulation depends on the orientation of the cortical neurons. L5 pyramidal neurons in the anterior and posterior wall of the precentral gyrus, which are located along the current direction are more entrained

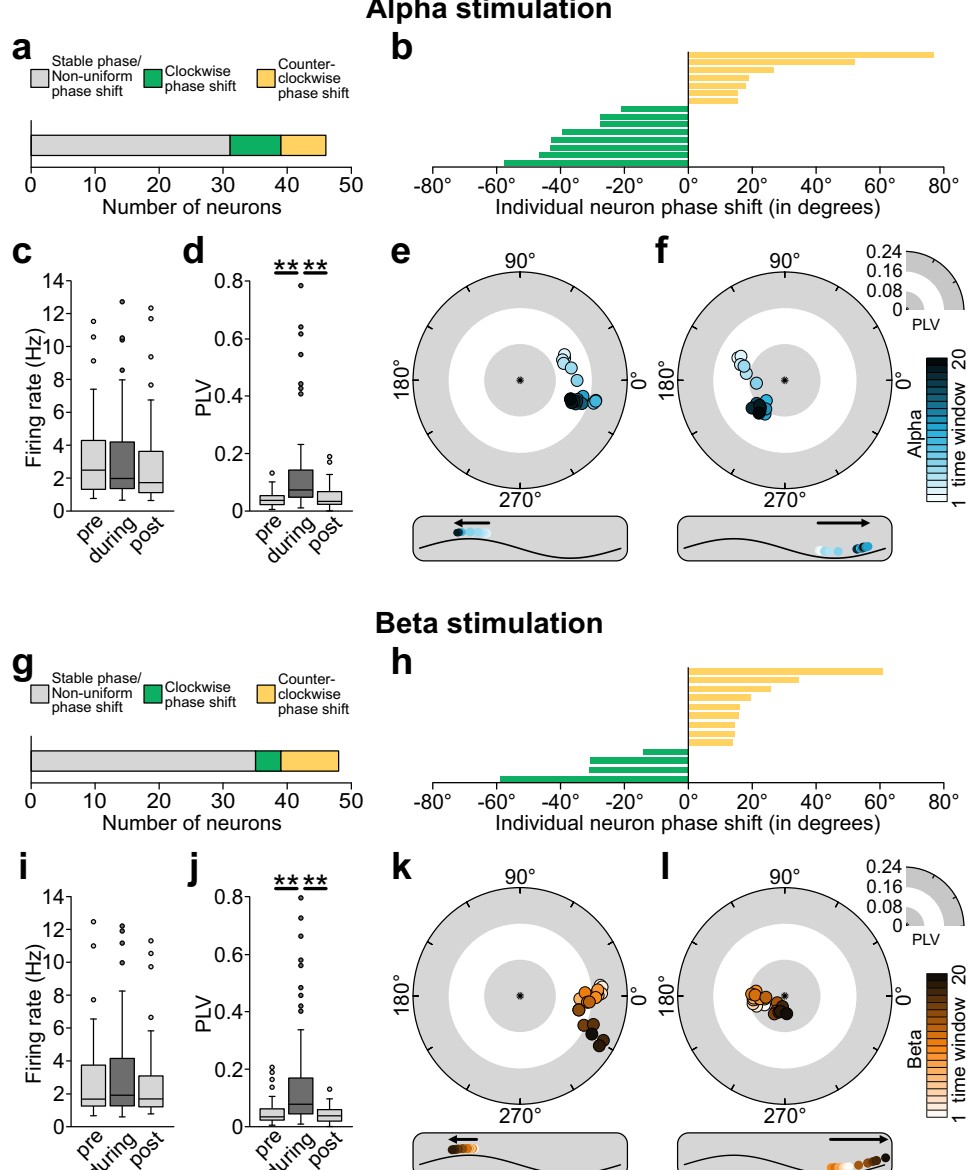

**Fig. 3 | Effects of AC stimulation on single unit activity in a non-human primate.** Results on stimulation at the alpha frequency (10 Hz) are shown in (**a**–**f**), and results on stimulation at beta frequency (20 Hz) are shown in (**g**–**l**). Shifts in preferred neural firing phase were calculated over time using a moving average (20 sliding windows) approach. **a** 46 of 81 recorded neurons showed significant entrainment during alpha stimulation. Of these 46 entrained neurons, 8 showed a clockwise (negative) phase shift over time, whereas 7 showed a counter-clockwise (positive) phase shift over time. Remaining neurons either showed stable entrainment or had phase shifts in more than one direction. **b** Representation of phase change (window 20 vs. window 1) of each neuron that showed a unidirectional shift during alpha stimulation. **c** Changes in average firing rate during AC stimulation, compared to pre- and post-stimulation ($n = 81$ neurons; paired $t$ test (two-sided); pre vs during: $t = 0.56$, $p = 0.578$, during vs post: $t = 1.44$, $p = 0.151$). Box limits represent 25th and 75th percentile, with the center line representing the median, and whiskers representing 1.5x interquartile range. **d** Changes in average phase locking value (PLV) during AC stimulation, compared to pre- and post-stimulation ($n = 81$ neurons; paired $t$ test (two-sided); pre vs during: $t = 4.93$, $p = 2.05e$-6, during vs post: $t = 4.56$, $p = 1.03e$-5). Note the significant increase during the stimulation compared

to before and after, suggesting significant entrainment. Box limits represent 25th and 75th percentile, with the center line representing the median, and whiskers representing 1.5x interquartile range. **e**, **f** Two example neurons showing a clockwise and counter-clockwise phase shift during alpha stimulation. **g** 48 neurons showed significant entrainment during beta stimulation. Of these, 4 showed a clockwise phase shift over time, whereas 9 showed a counter-clockwise phase shift over time. Remaining neurons either showed stable entrainment or had phase shifts in more than one direction. **h** Representation of phase change (window 20 vs. window 1) of each neuron that showed a unidirectional shift during beta stimulation. **i**, **j** Changes in average firing rate ($n = 81$ neurons; paired $t$ test (two-sided); pre vs during: $t = 1.23$, $p = 0.220$, during vs post: $t = 1.27$, $p = 0.205$) and PLV during AC stimulation ($n = 81$ neurons; paired $t$ test (two-sided); pre vs during: $t = 4.68$, $p = 6.09e$-6, during vs post: $t = 5.29$, $p = 4.04e$-7), compared to pre- and post-stimulation. Box limits represent 25th and 75th percentile, with the center line representing the median, and whiskers representing 1.5x interquartile range. **k**, **l** Two example neurons showing a clockwise and counter-clockwise phase shift during beta stimulation. *$p < 0.05$, **$p < 0.005$. Source data are provided as a Source Data file.

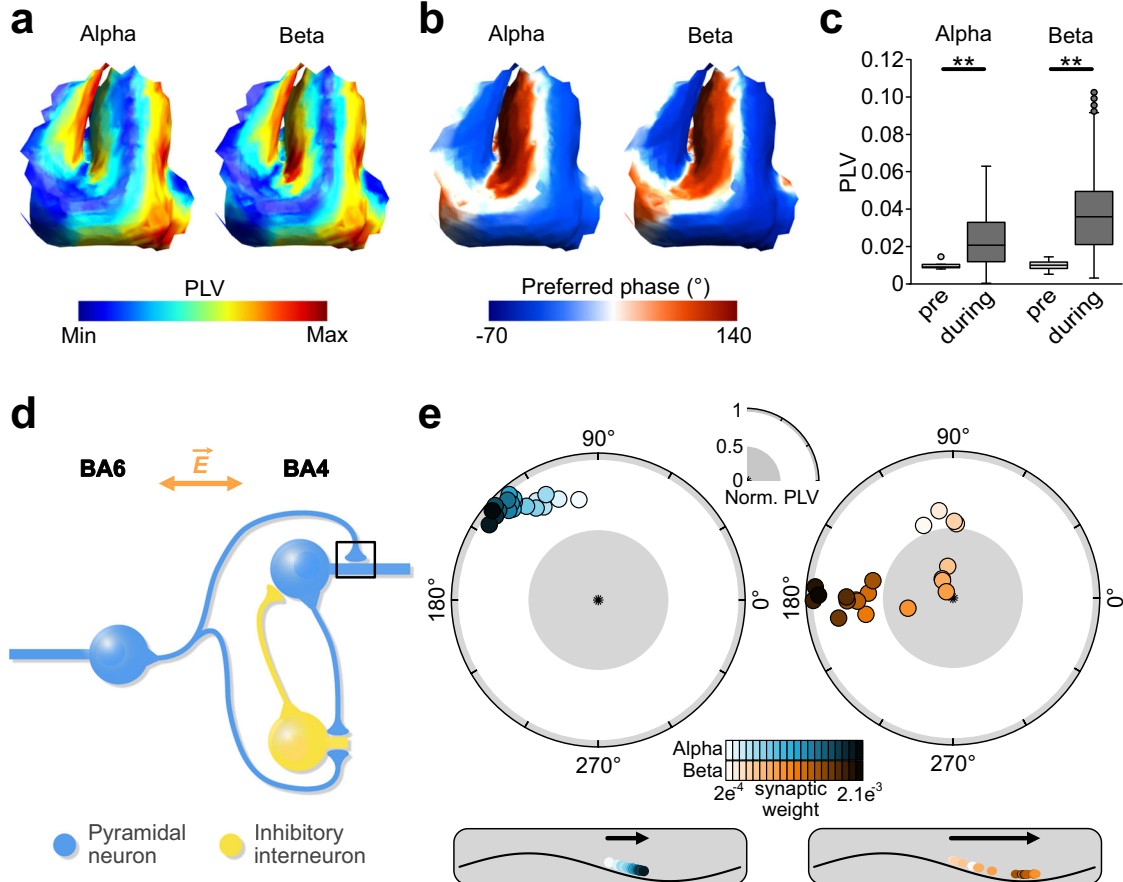

**Fig. 4 | Computational modeling of AC stimulation on single neuron activity.** In experiment 3 A, a ROI containing the precentral gyrus was populated by 4650 realistic L5 pyramidal neurons. **a** A spatial representation of phase locking values (PLV) during alpha and beta stimulation in the precentral gyrus. Using the current AC stimulation montage, strongest entrainment is observed in the walls of the precentral and central sulcus. **b** A spatial representation of preferred phase during alpha and beta stimulation. The anterior wall is entrained at phases between 90° and 180°, whereas the posterior wall is entrained at phases between −90° and 0°. **c** Change in PLV from no stimulation (baseline) compared to AC stimulation for alpha and beta stimulation (4650 realistic neurons). PLV was significantly increased (paired t-tests (two-sided); alpha: $t = 64.73$, $p < 1.0e-10$; beta: $t = 92.35$, $p < 1.0e-10$). In experiment 3B we used a microcircuit model consisting of two-compartment neurons to investigate whether AC stimulation-induced phase changes can be explained by NMDA synaptic plasticity. Box limits represent 25th and 75th percentile, with the center line representing the median, and whiskers representing 1.5x interquartile range. **d** A representation of the microcircuit model, consisting of a BA6-PY, a BA4-PY and a BA4-IN. **e** Systematic increases in synaptic weight of the BA6-PY to BA4-PY connection caused a gradual change in phase preference from -90° to -150° when alpha AC stimulation was applied. Similarly, beta AC stimulation and increased synaptic weight of the BA6-PY to BA4-PY connection resulted in a phase shift from -100° to -180°. These findings are consistent with findings in experiment 1, suggesting that tACS-related changes in phase preference of cortical motor output could result from induced synaptic plasticity. Source data are provided as a Source Data file.

than those at the crown and the bottom of the sulcus (Fig. 4a). Furthermore, L5 pyramidal neurons located in BA6 have a higher probability of spiking between at AC phases of 90° and 180°, whereas neurons located in BA4 are entrained between −90° and 0° (Fig. 4b). We repeated the analyses with intrinsic firing rates as observed in the baseline measurements of experiment 2 (3.02 spikes per second and 2.74 spikes per second for the alpha and beta blocks respectively). The results of these analyses were similar to the findings mentioned above (Supplementary Fig. S28).

### Experiment 3b: Phase-dependency of neural entrainment using neuronal microcircuit modeling

In experiment 3b, we used microcircuit modeling with two-compartment pyramidal (PY) and inhibitory (IN) neurons to replicate the phase shifts as observed in experiment 1. The physiology of the precentral gyrus was used as a template for developing our model (Fig. 4d). Our simplified model consists of a PY neuron in the AP axis, mimicking a premotor neuron in BA6. Additionally, a PY neuron in the PA axis and an IN neuron mimic primary motor neurons in BA4. Two alternative models are explored in Supplementary Fig. 29. We

then quantified neural entrainment and the preferential phase of neural firing with respect to the tACS oscillation. We hypothesized that phase shifts over time, as observed in experiment 1, result from tACS-induced synaptic plasticity. Therefore, in the model we systematically changed the N-methyl-D-aspartate (NMDA) synaptic strength of BA6-PY to BA4-PY connection. We found that BA4-PY showed a counter-clockwise phase shift for both alpha and beta stimulation conditions (Fig. 4e). The resulting trajectories (from -90° to -180°) in the model are in good agreement with phase shifts observed in experiment 1 (Fig. 2b). Additionally, we found similar phase shifts when increasing synaptic strength of α-amino-3-hydroxy-5-methyl-4-isoxazolepropionic acid (AMPA) for alpha stimulation (Supplementary Fig. 30A). For beta stimulation AMPA strength-related phase shifts were slightly weaker compared to NMDA. Increasing strength of γ-Aminobutyric acid-A ($GABA_A$) synaptic connections did not cause any phase shifts (Supplementary Fig. 30B). Together, this suggests that AMPA- and NMDA-mediated synaptic plasticity, but not $GABA_A$-mediated plasticity, between premotor and primary motor neurons captures the observed shifts in preferred phase of cortical motor output.

## Discussion

In three experiments involving in-vivo human and NHP data as well as computational modeling, we show that non-invasive neuromodulation, through the application of AC, biases timing of cortical excitability and induces phase precession. At the macroscopic scale in humans, we observed that cortical motor output was increased during particular phases of AC stimulation. Furthermore, gradual phase shifts were observed over stimulation blocks. Consistent with this observation, at the microscopic level in NHP, we show that a subset of entrained neurons displays phase precession in clockwise or counter-clockwise direction when AC stimulation is applied. Finally, under the hypothesis that phase precession underlies synaptic plasticity, we used multiscale computational modeling to investigate changing synaptic weights in a local network. We found that increases in synaptic plasticity in the model go hand in hand with AC stimulation-induced phase precession in experimental data.

Results in the non-human primate suggested that a subset of neurons displayed phase precession during the presence of an external AC field. Effects were similar for alpha and beta stimulation frequencies. Specifically, ~55–60% showed significant entrainment in at least half of the time windows used for analysis. This observation is consistent with previous NHP studies[39–41]. Furthermore, a subset of the entrained neurons showed a clockwise or counter-clockwise phase shift (~17% of neurons). It should be noted that several neurons showed non-uniform phase shifts (Supplementary Figs. S20B, S21B). A previous study has shown that tACS can entrain neurons at a phase that is different from their intrinsic preferred phase, hinting at the possibility of external electric field induced phase shifts[39]. However, to our knowledge, our study is the first to show AC-induced gradual phase precession. Also, our study provides further evidence that phase precession is not limited to the hippocampus and entorhinal cortex[18,23]. The intracranial recordings in the NHP spanned over a variety of regions in the frontal cortex (motor to prefrontal), suggesting that phase precession is a global process the brain[20]. Furthermore, phase precession was independent of firing rate, as it did not significantly change with stimulation, which is consistent with prior reports[40,41].

Reorganization of spike timing in relation to ongoing LFPs is thought to be one key mechanism for synaptic plasticity[28–30]. Therefore, it is expected that phase precession is not a phenomenon in isolated single cells, but rather reflects local network connectivity[20,30]. We explored whether AC stimulation-induced phase precession can be observed on a macroscopic scale in humans. For this, we probed corticospinal excitability by non-invasively activating local motor cortical networks with TMS. We used a previously verified algorithm[64,65] that allowed us for phase-specific probing of motor output in real-time. Overall, we observed that the phase of AC stimulation significantly modulated the magnitude of cortical output. At the preferred phase (between 90° and 180°) motor cortex excitability was ~7% larger than at the least preferred phase (~0°). This effect was independent of whether stimulation was applied at the alpha or beta frequency. Notably, this phase preference shifted over time. Although the exact phase shift trajectories differed between alpha and beta tACS, generally a shift from 90° to 180° was observed. Interestingly, in our study the 180° phase corresponds to an AP current direction. Based on the orientations of pyramidal neurons in the motor cortex[66], this implies that rhythmic depolarization of the premotor cortex (BA6) is associated with larger muscle responses, which is in line with prior observations[67,68]. BA6 neurons predominantly have disynaptic and trisynaptic connections to spinal motoneurons, in contrast to the monosynaptic connections of BA4[66,69–71]. Therefore, it is most likely that our observation of AC modulation reflects a transsynaptic circuit level effect of BA6 and BA4 neurons[72,73], which is in line with the idea that phase precession reflects plasticity local networks[20,30]. Together, our results in human participants indicated that macroscale phase precession relates to premotor-to-motor cortex connectivity.

Next, we explored the idea that synaptic plasticity is the underlying mechanism for network-level phase precession by using multiscale modeling. First, a model of the precentral gyrus with realistic layer 5 pyramidal cell was used to determine the neurons' preferred phase for depolarization. In line with our hypotheses, AC in PA direction depolarized BA4 neurons, whereas AC in AP direction depolarized BA6 neurons. Next, we used a microcircuit model with two-compartment PY and inhibitory IN neurons, based on simplified precentral anatomy[74,75]. We hypothesized that the observed phase precession in human cortical output is explained by changes in synaptic weight of the BA6-to-BA4 connection. In line with our expectations, we found that systematic increase in BA6-to-BA4 excitatory synaptic weights (NMDA and AMPA) mirrored the AC-induced phase shifts over time (Figs. 4e and 2b respectively). Also note that this phase precession is independent of firing rate, as AC stimulation did have no effect on the amount of spiking (Supplementary Fig. S27). Thus, our findings indicate that when tACS is applied to the motor region, counter-clockwise phase precession reflects synaptic plasticity in local excitatory premotor-to-motor connections. Changing inhibitory connection weights ($GABA_A$) had no effect, suggesting that inhibitory plasticity does not directly cause phase precession (Supplementary Fig. S30). Still, inhibitory neurons play a crucial role in overall network dynamics. In two alternative models, which did not contain GABAergic connections, phase precession direction and trajectory were significantly altered (Supplementary Fig. S29). This hints towards phase precession being network-dependent, which is in line with the various phase shift trajectories observed in different neurons of the NHP data. An exhaustive understanding of network dynamics on phase precession requires systematic testing of complex models with a variety of neuron types and layer-specific connections, which is beyond the scope of the present study.

Further evidence for the presence of short-term plastic effects comes from the observation that the cortical output increased over the course of a stimulation block (Supplementary Figs. S9, S10). Note that this cannot be explained by an accumulation effect of the TMS probe or an effect over time, as no increased responses were observed in a control condition without AC stimulation. Excitability returned to baseline before the start of each subsequent block, suggesting that there were no long-lasting aftereffects. This is likely explained by the relatively short stimulation duration of 6 min. In contrast, other studies have found that longer stimulation durations of 20 min can results in tACS aftereffects of an hour or more[59,76].

Since the present study is the first to show AC stimulation-induced phase precession, follow up questions arise that may inspire future research. Our study focused on the mechanistic understanding of phase precession, yet it would be fascinating to study the functional consequences of the observed effects. One hypothesis is the signaling between premotor and primary motor regions becomes more efficient. Premotor-to-motor connectivity is crucial for motor learning[77,78] and is abnormal in disorders of motor control[79,80]. Furthermore, tACS to motor regions has shown to facilitate motor learning[81–83]. While we observed phase precession in stimulation periods of approximately 6 min, it is possible that some neurons respond slower. Therefore, stimulation durations should be systematically investigated. Another avenue for forthcoming studies is to investigate phase precession in larger samples, as well as in other brain regions. The prefrontal cortex is associated with various cognitive functions and abnormal plasticity in this region relates to psychiatric disorders[84]. As such, investigating AC-induced phase precession in the prefrontal cortex, by for example investigating TMS-evoked potentials, can be of clinical importance[85]. In tandem with exploring effects in other brain areas, computational modeling of electric fields could be used to investigate different stimulation montages. Thereby, the effects of AC stimulation on different neural orientations and cell types can be studied. Additionally, systematically exploring various intensities would allow for establishing a

dose-response curve for externally induced phase precession[43,86]. It should be noted that using TMS only a finite number of phases can be probed (every 90° in this study). Future studies with next generation closed-loop TMS systems could investigate the phase precession in more resolution by testing oscillatory phases at finer grade, such as every 30°. Finally, the translation from single cell phase precession to network-level phase shifts may be explored in further detail. One way would be via combining intracortical and non-invasive recordings. Another possibility comes with the advancement of multiscale computational modeling. TMS, which was used to probe cortical excitability in this study, generates a complex cascade of direct- and indirect-waves[87]. As such, integrating modeling of AC fields and TMS dynamics would further enhance the understanding network-level phase precession.

In sum, across three studies, involving human, NHP and computational modeling data we show that: First, phase precession reflects both a single cell and a network-level process. This implies that phase precession operates on functional systems and may be a crucial mechanism for explaining human behavior, learning and cognition. Second, phase precession can be induced by the application of exogenous AC fields. Third, AC stimulation-induced phase precession directly relates to increase synaptic plasticity within local cortical connections. The latter two points demonstrate the therapeutic potential of applying external AC stimulation. Recent years have seen a significant increase in clinical trials using tACS[44]. Although still in an early stage, application of tACS has shown promising results in reducing symptoms of depression[88], Alzheimer's disease[89], and Parkinson's disease[90]. Altogether, the present study demonstrated that the shifting of preferred phase is one key mechanism by which tACS modulates neural activity.

## Methods

The present study consisted of three experiments. The first experiment was performed in healthy human volunteers. The second experiment was performed on a non-human primate (NHP). The final experiment consisted of multiscale computational modeling. An overview of experimental design and parameters is shown in Fig. 1a–c and Supplementary Table 1.

### Experiment 1 – electric fields on human neurophysiology
**Subjects.** We included 20 healthy volunteers (9 female, mean ± standard deviation age: 22.5 y ± 4.2) in a double-blind randomized crossover study. Each participant visited for two sessions (targeting mu and beta oscillations). All participants were between 18 and 45 years of age, right-handed, and without a history of (I) epilepsy of seizures, (II) neurological or psychiatric disorders, (III) head injuries, or (IV) metal or electric implants in the head, neck, or chest area. Besides these criteria, we used no pre-selection based on electrophysiological characteristics, such as motor threshold or sensorimotor oscillatory power. All volunteers gave written informed consent prior to participation. The study was approved by the institutional review board of the University of Minnesota.

**Electric stimulation.** A sinusoidal electric current was applied through two surface electrodes placed on the scalp targeting underlying cortical neurons, referred to as transcranial alternating current stimulation (tACS). The target location was the subject-specific finger (specifically first dorsal interosseous) region of the left primary motor cortex. TACS electrodes were placed 7 cm anterior and posterior of muscle hotspot, placed such that the electric field direction is approximately perpendicular to the orientation of the precentral gyrus[67]. Therefore, tACS-induced electric fields target distinct neural populations at different phases of the alternating current (AC). These phases were defined with respect to the posterior electrical stimulation electrode. Accordingly, 0° corresponds to the AC phase when the

electric field is aligned in posterior-to-anterior (PA) direction. In this phase the AC depolarizes soma in Brodmann area 4a and 4p located in the posterior bank of the precentral gyrus. A phase of 180° reflects an anterior-to-posterior (AP) electric field. During this phase AC depolarizes soma in Brodmann area 6 (premotor cortex; PMC) in the anterior precentral gyrus. Oscillation phases of 90° and 270° reflect the build up towards AP and PA current directions respectively.

Stimulation was applied with a multi-channel tACS StarStim 8 system (Neuroelectrics®, Cambridge, MA) through circular Ag/AgCl electrodes with 1 cm radius (Pistim; 3.14 cm²). Electrodes were directly attached to the scalp using adhesive conductive gel (Ten20, Weaver and Company, Colorado, USA) and impedances were kept below 10 kΩ. Two AC frequencies were used in separate sessions, based on individual peak activity in a three-minute resting-state EEG (Supplementary Fig. S2). For alpha stimulation peak activation between 7 and 13 Hz was used (mean ± standard error of mean, 9.92 ± 0.25). For beta tACS peak activation between 14 and 30 Hz was used (mean ± standard error of mean, 20.24 ± 0.89). The order of sessions was randomized. A stimulation intensity of 2 mA peak-to-peak was used. A ramp up and ramp down of 10 s was used. Per session four stimulation blocks of ~6 min (total stimulation time ~24 min) with ~2 min breaks in between were done.

**Probing cortical excitability.** To assess the excitability of the sensorimotor region we used single-pulse suprathreshold transcranial magnetic stimulation (TMS; Magventure, MagPro X100, Farum, Denmark), using a Cool-B65 figure-of-eight coil. Through the application of a short magnetic pulse an electric field is generated in the targeted brain region[91]. TMS was applied to the left primary motor cortex region that corresponds to the right hand first dorsal interosseous muscle. TMS-induced motor-evoked potentials were measured at the muscle using self-adhesive, disposable electrodes connected to electromyography[92,93]. For this, the sampling rate was set to 10 kHz using a BIOPAC ERS100C amplifier (BIOPAC systems, Inc., Goleta, CA, USA).

For TMS a 70 mm figure-of-eight coil was used and single biphasic pulse (280 μs) was applied at an intensity of 120% of the resting motor threshold. Motor threshold was defined as the minimal intensity to evoke a response at the target location, determined using an adaptive threshold-hunting algorithm[94,95]. A total of 600 TMS pulses were applied over four blocks of 150 trials (~6 min) concurrently with tACS. Inter-stimulus intervals were jittered between 2 and 3 s. TMS pulses were applied at four phases (0°, 90°, 180°, and 270°) of the alternating current stimulation cycle. To optimally align TMS to the tACS phase, we used a closed-loop system that reads out the oscillation phase in real time and simultaneously sends a trigger to the TMS[64,65]. Algorithm details and accuracy of phase targeting are shown in Supplementary Figs. S1 and S2.

**FEM head model.** FEM simulations of electric field distributions were done using SimNIBS version 3.2[96]. Electrode shape (circular), size (3.14 cm²), material (Ag/AgCl) and location (7 cm anterior and posterior of the motor hotspot), as well as intensity (2 mA peak-to-peak) were modeled in accordance with experimental procedures. Simulations were performed in an individual head model of a healthy adult male provided by SimNIBS ("Ernie"). Realistic conductivity values of different tissue types were used: $\sigma_{skin} = 0.465$ S/m, $\sigma_{bone} = 0.01$ S/m, $\sigma_{CSF} = 1.654$ S/m, $\sigma_{graymatter} = 0.275$ S/m, and $\sigma_{whitematter} = 0.126$ S/m (Windhoff et al., 2013). Gray matter volume was extracted for calculation of electric field strength (total volume: 1332 cm³).

**Data analysis.** First, an analysis on general AC phase-dependency of motor cortical output (measured by motor-evoked potentials) was performed. For this data from the four stimulation blocks was concatenated and motor-evoked potentials were averaged per stimulation

phase for both alpha and beta sessions. A repeated measures analysis of variance was conducted with stimulation phase (0°, 90°, 180°, 270°) and stimulation frequency (alpha, beta) as independent variables. Motor-evoked potential size was the dependent variable. Bonferroni-corrected t-test were performed as post-hoc analysis comparing phase conditions.

Second, we investigated whether the phase preference of motor cortical output changes over time. For this we calculated the polar vector strength and direction of the average normalized cortical excitability over phases. Specifically, x-coordinates were obtained from normalized motor-evoked potential size at 180° minus size at 0°, and y-coordinates were gathered from normalized motor-evoked potential size at 270° minus size at 90°. We assumed that motor-evoked potential data are distributed along the phase of targeted brain oscillation approximating von Mises probability density function (a "circular normal distribution"). This results in an estimate of phase preference and non-uniformity, where larger deviations from zero relate to a stronger bias towards a specific phase. This approach was repeated for 20 time windows using a moving average (sliding window length: 55 trials, 20 steps of 5 trials, averaged over four blocks). Subsequently, circular-linear correlations were calculated using the MATLAB circular statistics toolbox[97].

Finally, linear Pearson correlations were performed on motor-evoked potential amplitude regardless of phase, to investigate phase-independent changes over time. All analyses for human data were performed in JASP V14.0 (JASP Team, Amsterdam, The Netherlands) and MATLAB 2020b/2021b (MathWorks, Natick, MA) using customs scripts.

## Experiment 2 – electric fields on NHP neurophysiology

**Subject.** Data used in this study were collected from one rhesus macaque (Macaca mulatta, 13.5 kg, 9 years old, male). The animal was fitted with a cranial form-fitted chamber (Gray Matter Research, Bozeman, MT) and implanted with a 128-channel microdrive recording system (Gray Matter Research) over the left hemisphere. All animal procedures described here were approved by the Institutional Animal Care and Use Committee of the University of Minnesota (IACUC) and were conducted in accordance with the Public Health Service policy on Humane Care and Use of Laboratory Animals.

**Animal preparation and surgical procedure.** A high resolution T1 and T2-weighted MRI of the animal head were acquired at 10.5T[98,99] to create a 3D volume of the head to optimize the implantation of the microdrive and the head holder (Gray Matter Research, Bozeman, MT). A high-resolution CT image was additionally acquired to aid bone segmentation. The surgical procedures are divided into three separate surgeries: chamber implantation, craniotomy and microdrive implantation. The animal was stabilized with a titanium head holder and the 128-channel microdrive recording system was implanted in the region of interest. The recording system is composed of 128 individually movable glass-coated tungsten electrodes. Microdrive depths can be manually controlled using miniature screw-driven actuators. The chamber placement and microdrive positions were confirmed using post-operative magnetic resonance imaging (MRI) as well as computed tomography images (CT).

**Electrical stimulation.** AC stimulation was applied using the same apparatus as in experiment 1. Electrodes were positioned at the frontal and parieto-occipital cortex, roughly corresponding to FP1 and PO3 in the human 10–20 coordinates. For stimulation at the alpha and beta rhythm stimulation frequencies of 10 Hz and 20 Hz were used respectively. For each condition, we have four stimulation blocks of 6 min each separated by a 3-min period representing a total of 8 blocks in a single session. Before and after the stimulation blocks, we recorded a ~3-min period without stimulation. The two conditions were separated by a period of 25 min. Stimulation intensity was 2 mA (peak-to-peak), and a 10-s ramp up and down was used.

**Electrophysiological recordings.** The data was recorded while the animal sat comfortably in the chair without performing any type of task. No sensory stimulation of any kind was experimentally induced. The lights in the recordings room were on. Recordings were made using a 128-channel headstage (SpikeGadgets, San Francisco, CA, USA) with a sampling rate of 30,000 Hz and a bit depth of 16 bit. Electrodes were located in motor and prefrontal regions (Supplementary Fig. 13). Data was acquired using the dedicated open-source, cross-platform software Trodes. During the whole session, the animal was remotely monitored to ensure no signs of distress. The animal remained calm and did not show any sign of distress or anxiety.

**Data preprocessing.** Oscillating electrical stimulation can generate high amplitude artifacts in the raw recordings which, if not well removed, can affect spike sorting results. To address them, we computed the power spectral density of each channel and used spectral interpolation using FieldTrip MATLAB toolbox[100] to remove frequencies at which artifacts occur. Additionally, removing any frequency artifacts can help reduce undesired effects such as amplitude modulation. After the signal preprocessing, the channels were visually verified, and the power spectral density was computed again.

**Spike sorting.** Spike sorting was done using the open source MATLAB software Wave_Clus, an unsupervised spike sorting method[101]. Single units were first identified by bandpass filtering the raw preprocessed recordings (4th order Butterworth filter, 300–3000 Hz). An amplitude thresholding method based on the noise level was then applied (Eq. 1):

$$th = \lambda \times \sigma_n, \sigma_n = median\left\{\frac{|x|}{0.6745}\right\} \tag{1}$$

is the bandpass-filtered signal, $\lambda$ is a factor chosen by the user, $\sigma_n$ is an estimate of the standard deviation of the background noise[102]. In our case, the factor value of 4 was chosen. The higher it is, the more selective the method is. 32 data points are saved for each spike (i.e., ~1.25 ms), 8 samples before the negative or positive peak and 24 samples after and all spikes were aligned with the negative or positive peak. After spikes detection, the algorithm computes the wavelet transform of the spike shapes to extract features which are used as inputs for the clustering method. For further details, see[101]. The latter uses super-paramagnetic clustering, a method from statistical mechanics and based on nearest-neighbor's interactions. We used several metrics to verify each putative single unit: (1) the waveform shape and its variance, (2) the stability of spike amplitude and (3) the interstimulus interval distribution of each cluster. The signal-to-noise ratio for each single unit was computed using the following formula (Eq. 2)[103]:

$$SNR = \frac{1}{n_c}\sum_{i=1}^{n_c}\frac{\max(s_i) - \min(s_i)}{2 \times std(\varepsilon_i)} \tag{2}$$

Where $s_i$ is a vector of waveforms of spike $i$, $n_c$ is the total number of spikes of cluster $c$, and $\varepsilon_i = s_i - \bar{s}$ is the noise, $\bar{s}$ is the average spike waveform of cluster $c$. The higher the SNR is, the higher the quality of the clustering is. Each cluster was manually reviewed for quality control. Overall SNR, as well as examples of extracellular spike waveforms are shown in Supplementary Fig. 15.

**Data analysis.** We quantified the neurons firing activity by computing the phase-locking value (PLV) which estimates spike timing synchronization relative to an oscillation, in our case the tACS waveform. This metric is commonly used allowing a direct comparison with other

studies. The PLV was computed using the following formula (Eq. 3)[104]:

$$PLV = \left| \frac{\sum_{k=1}^{N} e^{i\theta_k}}{N} \right| \tag{3}$$

Where N is the number of action potentials and $\theta_k$ is the phase of the tACS stimulation at which the kth action potential occurs. A PLV of 0 means that there is no synchronization while a value of 1 means perfect synchronization. In addition, we used polar histograms—also called phase histograms—to highlight the preferred direction at which a neuron fires a spike (0 degrees equals the stimulation peak, 180 degrees equals the stimulation trough). The tACS waveform was based on the LFP artifact, we quantified the PLV and neuronal phase shift using a filtered version of the LFP. The latter was filtered with a 2nd-order Butterworth filter between 9 and 11 Hz for alpha stimulation and 19–21 Hz for beta stimulation. For the no stimulation period we computed the PLV and neuronal phase shift with regards to a virtual tACS waveform with the corresponding block frequency (10 Hz or 20 Hz, for alpha and beta stimulation respectively).

To classify neuron behaviors based on their phase shifts, we created a framework that could be divided into three main steps (Supplementary Fig. S16). Firstly, responsive neurons to tACS were identified based on a Rayleigh uniformity test ($p < 0.05$) using the MATLAB circular statistics toolbox[97]. We only kept responsive neurons in at least 10 time windows (or 50% of the total number of time windows). Since working with a circular data set can be challenging, we then computed the correlation coefficient between circular and linear variables using a specific function from the toolbox previously cited. We only kept neurons showing a high correlation coefficient (we chose 0.5 as a threshold value) and significance ($p < 0.05$). Finally, we kept neurons exhibiting a phase shift greater than 15° between the preferred phase in the first time window and the last time window.

All analyses on NHP data were performed in MATLAB using customs scripts on a regular desktop workstation.

## Experiment 3 – electric fields on microcircuit modeling of the motor cortex

### Experiment 3a - Populated head model with realistic neurons

**Neuron model.** We used multi-compartmental conductance-based neuron models with realistic morphologies from cortical layers generated in the NEURON v7.6[105]. NEURON is a simulation environment for simulating neurons with complex biophysical and anatomical properties. It is based on cable theory, which discretizes neuron morphology into small compartments for computing neural dynamics. The membrane voltage changes can be modeled along branches as a function of time and space. Branches are divided into small compartments of length dx, and the following equation can be numerically solved (Eq. 4):

$$\frac{1}{r_i}\frac{\partial^2 V(x,t)}{\partial x^2} - c_m \frac{\partial V(x,t)}{\partial t} + i_{ionic} = \frac{1}{r_i}\frac{\partial E(x,t)}{\partial x} \tag{4}$$

where $E_{\parallel}(x,t)$ is the induced electric field along the neuron compartment, $V(x,t)$ is the membrane potential, $r_i$ is the cytoplasmic resistance, $c_m$ is the cell membrane capacitance, $i_{ionic}$ are the currents passing through membrane ion channels, $x$ is the location of neuron compartment. The morphologically realistic neuron models were modified and adapted to the biophysical and anatomical properties of adult human cortical neurons[106]. Different ion channels and myelination are modeled as outlined in detail in[106]. Here, we used only L5 thick-tufted pyramidal cell models based on their hypothesized involvement in the TMS-evoked motor-evoked potential generation[66,69], and high responsiveness to the tACS in the modeling study[107].

**Synaptic activity modeling.** The synaptic current $I_{syn}$ at the a single postsynaptic compartment that results from a presynaptic spike was modeled as follows[108] (Eq. 5):

$$I_{syn}(t) = g_{syn}(t)\left(V(t) - E_{syn}\right) \tag{5}$$

where the effect of transmitters binding to and opening of postsynaptic receptors is a conductance change, $g_{syn}(t)$, in the postsynaptic membrane. $V(t)$ denotes the transmembrane potential of the postsynaptic neuron and $E_{syn}$ is the reversal potential of the ion channels that mediates the synaptic current. The dual exponential equation is used to describe the time course of the synaptic conductance[108] (Eq. 6):

$$g_{syn}(t) = \bar{g}_{syn} \frac{\tau_1 \tau_2}{\tau_1 - \tau_2}\left(\exp\left(-\frac{t - t_s}{\tau_1}\right) - \exp\left(-\frac{t - t_s}{\tau_2}\right)\right) \tag{6}$$

where $t_s$ is the time of a presynaptic spike, $\tau$ is the time constant of single exponential decay, $\tau_1$ and $\tau_2$ characterize the rise and fall times of the synaptic conductance in the dual exponential function. Like our previous modeling study[107], spiking activity of all neurons was generated through a synapse with a Poisson distribution with the same seed. The synaptic conductance resulting from the Poisson input was modeled with $E_{syn} = 0\,mV$, $\tau_1 = 2\,ms$ and $\tau_2 = 10\,ms$[109].

**Populating neurons in head model.** We can use the quasi-static approximation to separate the spatial and temporal components of the electric field[110,111]. For the spatial component, we calculated tACS-induced electric fields in the realistic volume conductor human head model using the head models implemented in SimNIBS[112]. A spherical region of interest with a radius of 11 mm was used on the head model to contain only the precentral gyrus. The gyrus was populated with single neuron models. The gray matter surface mesh of the modeled region contains 930 triangular elements. The L5 pyramidal neuron models are allocated in element by placing the soma at the center of the elements and a normalized depth of 0.65 between the gray and white matter surface meshes[106]. These model neurons were oriented so that their somatodendritic axes are normal to the gray matter surface. Five variants of L5 pyramidal neurons with different morphologies were co-located within each element, resulting in a total number of 4650 neurons in the region of interest. The placement of neurons and extraction of electric field vectors from the SimNIBS output were conducted in MATLAB. Because the aim of this investigation is to find the direct effect of electric fields on neurons, the neuron models were not interconnected. Therefore, higher electric fields are needed to achieve entrainment effects comparable to in vivo[107]. Electric fields on the cortex were scaled up by 5 resulting in a peak electric field strength of approximately 1.5 V/m.

**Coupling electric fields to neuron models.** The spatial component of the exogenous electric fields can be applied to cable models using the NEURON software's extracellular mechanism[105]. To couple the electric fields computed in the finite element method simulations of AC stimulation to the neuron models, the electric potential ($V_e$) was calculated for each cell by numerically integrating the electric field component along each neural compartment. In the case of uniformly distributed electric fields, the electric potential equation can be simplified as follows (Eq. 7)[113]:

$$V_e = -\int \vec{E} \cdot d\vec{l} = -\vec{E} \cdot \vec{l} = -(E_x x + E_y y + E_z z) \tag{7}$$

Where $\vec{E}$ is the electric field vector, $\vec{l}$ is the displacement. $E_x$, $E_y$ and $E_z$ indicate the Cartesian components of the electric field in a three-dimensional space, and x, y, and z denote the Cartesian coordinates of

each neuron compartment. The extracellular potential, $E_{\parallel}$ in Eq. 4, is calculated for each neuron compartment based on the external electric field induced by tACS using the electric potential. The electric field at the model compartments was interpolated from the mesoscopic tACS-induced electric fields computed in the model. Because the electric fields in our study are varying at low frequencies <100 Hz, they can be considered quasistatic and can be divided into spatial and temporal components[110,114]. After determining the spatial distribution of electric field, the electric field was computed by scaling the spatial distribution to the temporal component over time using a sinewave (10 or 20 Hz). The simulation duration was 9 min, involving a 3-min no stimulation baseline period, and a 6-min AC stimulation period. This setup matches the protocol of our presented human and animal experiments.

**Experiment 3b - Computational modeling of neural microcircuit Neuron models.** We hypothesized that synaptic connections between neurons may contribute to the phase-dependency of tACS effects and phase shifting of spiking activity observed in experiment 1 and 2. Here we aimed to investigate how the AC electric field entrains neurons that are coupled via synaptic connections in a microcircuit. The microcircuit model was composed of excitatory regular spiking pyramidal cells (PY) and a fast-spiking inhibitory interneuron (IN). We implemented two-compartment models of each neuron consisting of soma and a dendrite. The parameters used for the morphologies of the neurons are shown in Supplementary Table 3. We used NetPyNE[115], a Python package to simulate and model biological neuronal networks using the NEURON simulation environment[105]. The membrane potential of the model neuron was calculated in NEURON.

Active currents in PY include a fast sodium current ($I_{Na}$), fast potassium current ($I_{Kv}$), slow non-inactivating potassium current ($I_{km}$), leak current ($I_L$), calcium current ($I_{Ca}$), and Calcium-dependent potassium current ($I_{KCa}$). The kinetic equations of these currents are adapted from the model[116]. Membrane electrical properties, such as membrane capacitance and ion channel conductance of the model are modified to generate a regular-spiking firing pattern. The IN was adapted from single-compartment model[117] and a dendritic compartment is added to allow for electric field coupling. INs only contain ($I_{Na}$) and ($I_{Kv}$) currents to create fast-spiking activity.

**Modeling of AC stimulation.** We used a sinusoidal stimulation waveform and simulated the microcircuit at an electric field strength of 3 V/m. Two stimulation frequencies (10 Hz and 20 Hz) were applied during the simulation, consistent with our experimental conditions. The electric field was spatially uniform and aligned with the horizontal axis which corresponds to the somatodendritic axis of the model cells. The total duration of the stimulation was 10 min, with a 6-min no-stimulation baseline period and a 6-minute period with AC stimulation. We coupled the AC electric field to all two-compartment neuron models, as in experiment 3a.

**Synaptic current.** We implemented α-amino-3hydroxy-5-isoxazolepropionic acid (AMPA), N-methyl-D-aspartate (NMDA), and γ-aminobutyric acid type A (GABA$_A$) mediated synaptic current in the microcircuit model. Using the same dual exponential equation (Eq. 6), synaptic conductance resulting from excitatory (AMPA and NMDA) and inhibitory (GABA$_A$) synapses were simulated with parameters in Supplementary Table 4[118]. All neurons received Poisson random inputs to mimic subthreshold synaptic inputs from the background.

**Microcircuit connectivity.** The simplified microcircuit is composed of 2 PYs and 1 IN and all neurons were spatially arranged in a specific orientation that mimics the spatial distribution of neurons in the precentral gyrus (Fig. 4d). One PY corresponds to the premotor neuron in BA6, with the somatodendritic axis oriented normal to the anterior

gray matter surface. The other PY is oriented normal to the posterior gray matter surface, as well as the IN corresponding to the primary motor cortex neurons in BA4. Small INs can be orientated in any direction, and they are not very responsive to the electric field[107]. For simplicity, IN in the microcircuit has the same orientation as the BA4-PY. BA6-PY connects to the BA4-PY and IN through excitatory connections with a synaptic delay of 2 ms. This connection in the model emulates the premotor-to-motor projection. BA4-PY was connected to the IN locally with a reciprocal connection and a shorter synaptic delay. All excitatory synapses were connected to the dendritic compartment of the model neurons and the inhibitory synapse was connected to the soma. The microcircuit parameters are described in Supplementary Table 5. Several studies have suggested that tACS induced entrainment effect is stronger when the stimulation frequency is close to the intrinsic oscillations of neurons[38,63]. Therefore, the synaptic Poisson inputs to PYs were tuned to match each stimulation condition. For alpha stimulation, PYs are tuned to firing around 10 spike/second. For beta stimulation, PYs are tuned to fire at low beta rhythm (around 17 spikes/second). Due to the higher firing rate of BA6-PY during beta stimulation, BA6-PY-to-IN NMDA synaptic strength was reduced to keep the firing rate of IN consistent with the alpha stimulation condition.

**Modeling of plasticity.** Although direct monosynaptic output to muscles primarily originates in BA4a and BA4p, pyramidal neurons in BA6 play an important role in motor response generation via monosynaptic projections to BA4[66,75]. To replicate and explain the experimental results observed from experiment 1, we explored different realistic parameters in the model. We hypothesized that phase shifting of cortical motor excitability could be related to NMDA receptor-mediated synaptic plasticity during the stimulation period[59]. Long-term potentiation was modeled as persistent strengthening of the NMDA synaptic connection from BA6-PY to BA4-PY. For each stimulation frequency, we simulated 20 trials and systematically increased the NMDA weights with a constant increment while keeping the other parameters unchanged.

**Data analysis.** As in experiment 2, PLV was used to quantify the neural entrainment. For each neuron, the preferred phase of the neuron was calculated by taking the average of phases of spikes corresponding to the stimulation waveform. Data analyses for computational modeling were conducted in MATLAB.

### Reporting summary
Further information on research design is available in the Nature Portfolio Reporting Summary linked to this article.

## Data availability
Experimental human data is available upon a request to the principal investigator according to the IRB protocol approved by the University of Minnesota. Experimental animal data is available upon a request to the principal investigator according to the IACUC protocol approved by the University of Minnesota. Source data are provided with this paper.

## Code availability
Matlab version 2020b/2021b was used for data analysis. Matlab 2020b/2021b was used for statistical analysis using the statistics and machine learning toolbox. For NHP spiking data analysis open-source and freely available toolboxes Wave_clus and Fieldtrip were used. For computational modeling was done in Python using open-source and freely available toolboxes NetPyNE and NEURON. Further, simulations were run using our open-source NeMo-TMS repository (https://github.com/OpitzLab/NeMo-TMS) and the open-source repository related to Aberra et al. [106] (https://github.com/Aman-A/TMSsim_Aberra2019).

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

## Acknowledgements

Research presented here was supported by the National Institute of Health (NIH grants: R01EB034143, R01NS109498, R01EB031765, R01MH128177, RF1MH124909, P30NI048742, P41EB027061), Behavior and Brain Research Foundation (Young Investigator grant), and the University of Minnesota's MnDRIVE Initiative. Additionally, we acknowledge Minnesota Supercomputing Institute (MSI) for providing resources.

## Author contributions

MW: Conceptualization, Methodology, Data collection (experiment 1), Data analysis (experiment 1), Writing, Reviewing & editing. HT: Animal handling, Methodology, Data collection (experiment 2), Data analysis (experiment 2), Writing, Reviewing & editing. ZZ: Methodology, Model generation and analysis (experiment 3), Writing, Reviewing & editing. SS: Methodology, Development of hardware/software, Reviewing & editing. ZJH: Data collection (experiment 1), Participant management, Data handling. JR: Data collection (experiment 1), Participant management, Data handling. IA: Conceptualization, Methodology, Development of hardware/software, Reviewing & editing. NDP: Data analysis, Reviewing & editing. JZ: Resources, Animal handling, Supervision, Funding acquisition, Reviewing & editing. AO: Resources, Supervision, Project administration, Funding Acquisition, Conceptualization, Methodology, Writing, Reviewing & editing.

## Competing interests

The authors declare no competing interests.
