## [Peer Review File · Nature Communications]

Induced neural phase precession through exogenous electric fieldsREVIEWER COMMENTS

Reviewer #1 (Remarks to the Author):

Wischnewschi and Tran et al. describe phase precession under external AC fields in three combined studies, including humans, a macaque, and a computational model. The results are entirely novel and of high relevance, both for the tACS community and for neuroscience in general, as phase precession may be an important mechanism in various parts of the brain. I consider the presented data compelling, but I have some concerns that I would like to see addressed.

Major:

1) Comparability of the different experiments. As results and methods are slightly differently described for the human and animal part, it would be good to have an overview e.g. in the form of a table, to show where the experiments differed in settings and in the outcome.

a) Both humans and macaques were stimulated with 1mA peak-to-baseline. For the human part, the manuscript presents E-field simulations in a sample head model, but not for the macaque part. I would expect that the same current intensity leads to higher E-fields in the macaque, as their head is smaller and the skull thinner. Could the authors either add an E-field model for the macaque and compare the E-fields, or discuss the expected difference in E-fields?

b) Where were the 128 recording contacts in the macaque located? I tried to find this information in both results and methods, but only found that they were in the 'left hemisphere'. Did the recording cover the motor cortex? If yes, were the results in motor cortex any different from the other areas? If no, could the authors comment on comparability with the human data?

c) There seem to be faster/more pronounced phase shifts in the human data (~60-70 degrees in 6 min) compared to the macaque data (~30-40 degrees in 6 min), although the E-fields in humans are lower than in the macaque. Could the authors comment on this discrepancy?

d) I understand that the authors upscaled E-field strengths in the models to compensate for missing lateral connectivity. But why does the microscopic model receive double the field strength compared to the macroscopic model (3 V/m vs 1.5 V/m)?

2) 'Gradual' shifts in spike timing / phase precession.

a) The authors report that in the animal data, some neurons showed a phase drift in one direction, whereas other neurons showed bidirectional drifts. There seems to be a focus on describing unidirectional drifts as these indicate phase precession as known from hippocampal place cells and entorhinal grid cells. Similarly, there are smooth phase drifts in the human data. However, given that in the analysis, there was extensive smoothing over time for both the human and the macaque data (in the human data, segments of 55 trials with steps of 5 trials; in the macaque data, segments of 132 seconds with steps of 12 seconds [the manuscript says 'overlap of 12 seconds', but given that the total duration

of a block was 360s and there were 20 time windows, I assume that the authors meant the step length and not the overlap]): are these unidirectional drifts statistically significant? In other words, would randomly shuffled data after the same smoothing (same length of segments and of steps) lead to fewer (animal) / less pronounced (human) unidirectional drifts?

b) In the human study, only 4 phases were tested, with 90 degrees difference. If I understand correctly, the precise phase was estimated by averaging data of 55 trials. I see that it is practically impossible to test a large number of different phases. But could the authors comment on the validity of this approach to show drifts of 60-70 degrees (smaller than the 90 degree difference between the phases that were tested)?

c) The authors assume that there is a sort of 'reset' after each block, and thus data from the four blocks (for both humans and macaques) can be averaged. Is there evidence that the (plastic) state is actually reset after each break, e.g. by comparing preferred phases or other variables at the beginning of each block across the four blocks?

3) Group sizes. With 20 participants and one macaque, group sizes are a bit on the low side. I do not think that this is a serious problem here as the manuscript combines evidence from humans, animals, and a computational model, but it should be mentioned in the discussion.

Minor:

4) How was the 'tACS waveform' in the macaque study defined? Was it based on the LFP artifact or on the injected current?

5) In equation (7), the reader may get the impression that this is a surface integral and then get confused (at least I did). To make it easier for the reader, I would suggest calling the displacement l (instead of s , which is often used for surfaces), or directly indicating that it is a line integral.

6) In the computational model, the authors adjust the firing rate for α - and β -tACS. It is not clear to me why the oscillation frequency of the model must be equal to the firing rate of the neurons. Couldn't the neurons keep their firing rate, and only synchronization properties are adjusted to the α - or β -range, to avoid any bias due to different firing rates?

Bettina Schwab

Reviewer #2 (Remarks to the Author):

In this manuscript, Wischnewski, Tran, and their colleagues aim to establish a causal link between LFP dynamics and phase precession, as stated in the abstract. The authors employ a unique approach, combining human and monkey data with computational, multi-scale models.

I have three core comments that I believe need to be addressed in order to improve/strengthen the study:

A) In my opinion, the connection between the study's findings and physiological phase precession is not clear. I personally do not think that there is enough evidence to make specific inferences about physiological processes, considering that the phase precession discussed in the manuscript is relative to exogenous stimulation. At least, the authors should provide convincing reasons, for instance in the discussion, of why they think that the exogenous manipulations they applied may provide direct insights about physiological phase precession.

B) To enhance the robustness of the results, further analysis is necessary, particularly considering that several arbitrary thresholds are applied to the data for Experiment 2. A more thorough examination of those thresholds and their impact on the analysis outcomes would strengthen the results of this manuscript.

C) The computational model in the study should be utilized more extensively to elucidate the underlying mechanisms behind the reported effects. By using the model in a more comprehensive manner, the authors can provide a deeper understanding of the observed effects of stimulation on neuronal activity.

Below, you can find specific comments related to each of the previous points.

A.

1 - It appears that the paper's main focus is on the effects of exogenous electrical stimulation on brain activity, rather than directly on physiological phase precession. In particular, there is a lack of LFP recordings, which diminishes the direct association between LFP dynamics and the study's findings. In my opinion, the paper would benefit from emphasizing the effects of AC stimulation instead of attempting to establish connections to physiological processes.

2 - On a broader note, phase precession is typically believed to facilitate coding, learning, or plasticity (as also stated by the authors). However, since the subjects in this study are not actively engaged in any specific tasks or behaviors, it raises the question of what would be the purpose of phase precession in this situation. At least, further exploration and discussion of the potential functional significance in the absence of specific tasks would add depth to the study.

3 - Is it possible that the observed phase precession is merely a result of LFP entrainment? If a neuron maintains a fixed spiking phase relative to the LFP, and the LFP becomes progressively entrained by the stimulation, it could lead to an apparent shift in the neuron's phase relative to the stimulation. However, in this case, such a shift would not correspond to actual phase precession (the phase would be unchanged relative to the LFP). Are there any obvious reasons to exclude this hypothesis?

4 - The phase changes depicted in Fig. 2 appear rather abrupt and may not align with the typical definition of phase precession. Can they be called "phase precession"?

B.

5 - In Fig. 2, it is generally unclear whether the reported results are aggregated across subjects. Are the results consistent across subjects or are they driven by a few?

6 - Could the general increase in MEP size (Supplementary Fig. S5) potentially confound the reported results? For instance, is there any correlation between the phase shift and the slope of the MEP size increase across subjects?

7 - Is there a correlation between the MEP size and changes in the subjects' state? Is there, for instance, any systematic change in pupil size, as a proxy for arousal, which may explain the MEP size drift and/or maybe phase changes?

8 - Is Experiment 2 based on a single recording session?

9 - The methods section lacks information regarding the definition of PLV in the baseline. How is it computed, since there is no stimulation in that period?

10 - In line 489, it is mentioned that various metrics were utilized to assess the stability of the units; however, there is no explicit mention or reporting of the specific analyses conducted. If plots or additional information demonstrating the stability of the recordings and excluding recording drift are available, it would be beneficial to include them in the manuscript. Examining such plots or analyses

would help validate the reliability of the results and provide a clearer understanding of whether the observed phase changes are influenced by recording drift or other factors.

11 - Understanding the robustness of the phase change results in Experiment 2 is challenging due to the extensive pre-processing described in the Methods section (lines 511-518). It is crucial to determine if the observed phase shift, which is the main result derived from these experiments, remains robust when altering the thresholding parameters. How do the results change when these thresholds are modified? Additionally, I am uncertain about the application of the final threshold ($> 15^\circ$). Is the data analyzed for phase shifts limited to neurons that already exhibited a phase shift? If this is the case, it might be expected to find neurons with phase shifts. Could the authors clarify this point?

12 - Another related question arises regarding the application of a threshold on phase shifts. The data presented, such as in Figure 2B, appears to have gaps in the middle of the distribution due, I guess, to the applied threshold. If these missing data points were included, one might expect the entire distribution to appear more continuous. Hence, it raises the question of whether the observed phase shifts are genuine or merely a result of fluctuations. Is it possible to simulate a ground truth to shed light on this matter? For instance, by simulating a population of neurons with similar firing rate distributions as in the experimental data and for a similar amount of time, the authors could investigate what the distribution of phase shifts would look like and quantify the percentage of false-positives.

13 - Could the authors clarify the definition of a 'non-uniform' phase shift for neurons? What are the specific quantitative differences between neurons with non-uniform phase shifts and those exhibiting a 'uniform' phase shift? Additionally, is the finite duration of the experiment (relative to the firing rate distributions) a potential concern for this classification? Could it be possible, for instance, that neurons with phase shifts are defined as such just because they do not fire much and therefore it is easier to find periods where it seems like there is a uniform shift? Similarly, is it possible that some units with non-uniform phase shifts would simply experience their phase change at a later time? I also noticed that some neurons in Figures S7-8 exhibit similar time profiles to the examples shown in Figures S10-11. Could the authors provide further insight into this observation?

14 - As for Experiment 1, are there any signs that the animal's arousal state is stable during the stimulation period in Experiment 2?

C.

15 - In Experiment 3a the firing rate was set at 10 and 20 spikes per second. Why was the firing rate set like that? It would seem that a more reasonable choice would be to fix the rates to the ones recorded in Experiment 2.

16 - What could be the reason for observing phase precession that is independent of the LFP frequency, as indicated by the generic phase shift observed in many units for both alpha and beta stimulation in Figure S9? Why are the phases and phase shifts different for the two stimulation frequencies? Could the model be used to understand this observation?

17 - While the clarity of the results in Fig. 4A-C is evident, I think that this large scale simulation could be more valuable if the authors were estimating the relative contribution of multiple factors, such as electric field magnitude, direction, neuronal somatodendritic axis and specific cell examples (5), to the estimated PLV.

18 - Why are there neurons in the model for which the PLV decreases relative to baseline?

19 - I think that the model should be explored more to answer a specific set of questions. For instance, if the inputs are boosted, something that would provide an increased output that would mimic increased MEP size, does the phase change?

20 - While the hypothesis of NMDA plasticity is reasonable, it is worth considering that there might be multiple parameters that could potentially induce phase shifts. Exploring alternative possibilities could be beneficial if the authors aim to determine the primary candidate responsible for the observed phase shifts.

Minor comments

21 - The concluding sentence in the discussion, in its current form, seems better suited for an introduction or abstract rather than as the final sentence of the paper.

22 - Line 570: typo "electric fields on neuros"

23 - Incorrect reference in Supplementary Fig. S6. "Similar data as in Supplementary Fig. 2". I think it should be Supplementary Fig. 5.

REVIEWER COMMENTS

Reviewer #1 (Remarks to the Author):

Wischniewski and Tran et al. describe phase precession under external AC fields in three combined studies, including humans, a macaque, and a computational model. The results are entirely novel and of high relevance, both for the tACS community and for neuroscience in general, as phase precession may be an important mechanism in various parts of the brain. I consider the presented data compelling, but I have some concerns that I would like to see addressed.

Reply: First of all, we would like to thank the reviewer for the kind words and the insightful comments. We address these points in detail below and we believe that, thanks to the reviewer, the manuscript improved significantly.

In summary, we added an experimental overview table and the e-field simulation for the NHP. Also, we added a figure clarifying electrode locations and ran several control analyses.

Major:

1) Comparability of the different experiments. As results and methods are slightly differently described for the human and animal part, it would be good to have an overview e.g. in the form of a table, to show where the experiments differed in settings and in the outcome.

Reply: Thank you, this is a great suggestion. We added a supplementary table comparing the experimental parameters. We refer to the table in the methods section on page 17, lines 411-412:

“An overview of experimental design and parameters is shown in Fig. 1A-C and Supplementary Table 1.”

Supplementary Table 1. Overview of alternating current stimulation parameters and experimental design for the three experiments of this study.

	Experiment 1 (human)	Experiment 2 (NHP)	Experiment 3a (population modelling)	Experiment 3b (microcircuit modelling)
tACS intensity	1 mA	1 mA	-	-
Max. e-field	0.31 mV/mm	0.84 mV/mm	1.5 mV/mm	3 mV/mm
tACS duration	4 x 6 min (24 min total)	4 x 6 min (24 min total)	6 min	6 min
tACS frequency	On average, 9.92 Hz and 20.24 Hz	10 Hz and 20 Hz	10 Hz and 20 Hz	10 Hz and 20 Hz
tACS montage	2 PiStim (Ø 3.14 cm ²) electrodes 7 cm anterior and posterior of M1	2 PiStim (Ø 3.14 cm ²) electrodes Fp1 and PO3	-	-
Outcome measure	MEP	Single-unit activity	Single-unit activity	Single-unit activity

a) Both humans and macaques were stimulated with 1mA peak-to-baseline. For the human part, the manuscript presents E-field simulations in a sample head model, but not for the macaque part. I would expect that the same current intensity leads to higher E-fields in the macaque, as their head is smaller and the skull thinner. Could the authors either add an E-field model for the macaque and compare the E-fields, or discuss the expected difference in E-fields?

Reply: We added the e-field simulation for the NHP. As expected, the induced electric fields are higher (maximum e-field = 0.84 mV/mm) compared to the human. E-fields in the region that contained the microdrive electrodes varied between ~0.1 and ~0.75 mV/mm. We added the simulation to the supplementary materials (**Supplementary Fig. 13**) and refer to it on page 8, lines 161-163:

*“The electric field strength had a maximum value of 0.84 mV/mm and was between ~0.1 and ~0.75 mV/mm in the region that contained the recording electrodes (**Supplementary Fig. S13**).”*

Supplementary Fig. S13. Electric field distribution in the NHP brain using a tACS montage with electrodes placed at frontal and parieto-occipital locations (roughly corresponding to FP1 and PO3 in the human 10-20 coordinates).

b) Where were the 128 recording contacts in the macaque located? I tried to find this information in both results and methods, but only found that they were in the 'left hemisphere'. Did the recording cover the motor cortex? If yes, were the results in motor cortex any different from the other areas? If no, could the authors comment on comparability with the human data?

Reply: The recording electrodes were in a region that spans from the primary motor cortex to the prefrontal cortex. We included a figure with electrode locations in the supplementary data (**Supplementary fig. 12**), and refer to this figure on page 20, line 532: “Electrodes were located in motor and prefrontal regions (**Supplementary Fig. 12**).”

We did not observe regional biases and neurons that displayed clockwise, or counter-clockwise phase shifts were distributed throughout the area (please compare to Supplementary fig. 15).

Supplementary Fig. S12. Schematic of regional coverage of the 128-channel microdrive recording systems in the NHP. Note that electrode depth differs between electrodes.

c) There seem to be faster/more pronounced phase shifts in the human data (~60-70 degrees in 6 min) compared to the macaque data (~30-40 degrees in 6 min), although the E-fields in humans are lower than in the macaque. Could the authors comment on this discrepancy?

Reply: Interpreting the exact phase shifts is not trivial. However, one potential explanation relates to the network dynamics of the neuron that displays phase precession. On top of the model we used in experiment 3B, we investigated phase shifts in two additional simple network models. One with a pyramidal-pyramidal connection with opposing neuron orientation, and one with a pyramidal-pyramidal connection with the same neuron orientation. In contrast to the original model, the PY-PY model with opposing directions showed a clockwise phase shift. The PY-PY model with the same orientation showed only minor (counter-clockwise) phase shifts of $\sim 10^\circ$. Certainly, in reality connective properties are far more complex, but what this additional analysis shows is that the trajectory and degree of phase shifts heavily depends on the network. As such, the wide range of phase shifts ($\sim 15^\circ$ to 80°) and differing directions (clockwise and counter-clockwise) we observed in the different neurons may, in part, depend on their connections and neural orientations. Further, differences in brain structure (such as gyrification) and brain network dynamics between the humans and monkey may drive the faster/more pronounced phase shifts in humans. We added the information on the additional models to the supplementary materials (**Supplementary Fig. 26**). Furthermore, we refer to these results on page 11, line 246:

“Two alternative models are explored in **Supplementary Fig. 26**”

Supplementary Fig. S26. We explored the effects of two alternative simplified network models. A) a model with a pyramidal-pyramidal connection with opposing neuron orientations and without inhibitory interneuron. This would reflect an uninhibited connection from BA6 to BA4. In contrast to the original model in the main results, this model showed a clockwise phase shift with increasing synaptic weights, going from $\sim 240^\circ$ to $\sim 180^\circ$ for alpha stimulation, and going from $\sim 280^\circ$ to $\sim 230^\circ$ for beta stimulation. B) A model with a pyramidal-pyramidal connection with the same neuron orientations and without inhibitory interneuron. This would reflect an uninhibited connection within BA4. The results of this model showed a very small counter-clockwise phase shift of approximately 10° for both stimulation frequencies. Furthermore, the shift occurs at the opposite end of the oscillation cycle compared to the original model (Fig. 4E).

d) I understand that the authors upscaled E-field strengths in the models to compensate for missing lateral connectivity. But why does the microscopic model receive double the field strength compared to the macroscopic model (3 V/m vs 1.5 V/m)?

Reply: Generally, computational models have higher E-field threshold as they are simplified compared to in vivo. The nature of the two models used here is very different. In experiment 3A we used unconnected neurons with complex morphologies (such as complex dendrites and axon arbors) which have lower thresholds and generally more responsive to E-fields. For this model we used a maximum field strength of 1.5 V/m based on a previous study of our group¹. According to these results intensities higher than 1.5 V/m simply result in linear increases. The network model in experiment 3B uses three ball-and-stick neurons. For such simplified models, higher field strengths are typically used^{2,3}. Note that their purpose is to reflect the mechanism conceptually. Thus, the intensity used for this model is not reflective of actual stimulation intensities.

2) 'Gradual' shifts in spike timing / phase precession.

a) The authors report that in the animal data, some neurons showed a phase drift in one direction, whereas other neurons showed bidirectional drifts. There seems to be a focus on describing unidirectional drifts as these indicate phase precession as known from hippocampal place cells and entorhinal grid cells. Similarly, there are smooth phase drifts in the human data. However, given that in the analysis, there was extensive smoothing over time for both the human and the macaque data (in the human data, segments of 55 trials with steps of 5 trials; in the macaque data, segments of 132 seconds with steps of 12 seconds [the manuscript says 'overlap of 12 seconds', but given that the total duration of a block was 360s and there were 20 time windows, I assume that the authors meant the step length and not the overlap]): are these unidirectional drifts statistically significant? In other words, would randomly shuffled data after the same smoothing (same length of segments and of steps) lead to fewer (animal) / less pronounced (human) unidirectional drifts?

Reply: 12 seconds indeed refers to the step length, not the overlap. We clarified this in the manuscript.

In the human data we determined the likelihood of a "spurious" unidirectional phase shifts by performing a permutation test. For that we re-shuffled the MEP data and examined the unidirectional phase shifts 10,000 times. The results of this analysis suggest that the actual observed correlations characterizing the phase shifts (alpha $r = 0.655$, beta $r = 0.825$) fall in the 95th percentile of the permuted distributions (alpha $p = 0.041$, beta $p < 0.001$). This suggests that finding spurious unidirectional shifts in our data is unlikely ($p < 0.05$). We added this analysis to the supplementary data (Fig. S5) and refer to it in the main text on page 6, lines 127-128: "Permutation testing on excitability with randomized phases suggested that spurious uniform phase shifts in our data are unlikely ($p < 0.05$, Supplementary Fig. S5)."

Supplementary Fig. S5. Permutation testing on circular-linear correlations in experiment 1. Phase information of MEP was shuffled in 10000 permutations. Actual observed correlation values for alpha ($r = 0.655$) and beta ($r = 0.825$) both fall in the 95th percentile of the permutation distribution ($p = 0.041$ and $p > 0.001$, respectively), suggesting that the observed phase shifts in human data are unlikely to occur spuriously.

In the NHP data we compared phase preference during AC stimulation to a 6-minute no stimulation period. In the no stimulation period, the phase was determined with respect to a virtual tACS signal. The same thresholding steps were used for determining phase shifts in the no stimulation period (Supplementary Fig. S14). Of the 28 neurons (15 for alpha, 13 for beta) that displayed significant phase precession during tACS, none were found to display phase shifts in the no stimulation period. Examples are shown in Supplementary Fig. S20. We refer to these results on page 9, lines 194-197:

“Of the 28 neurons that showed significant phase precession during AC stimulation (alpha $n = 15$, beta $n = 13$), none were found to display significant phase shifts in a 6-minute period without stimulation (Supplementary Fig. S21).”

Supplementary Fig. S21. Phase shifts during alpha (blue) and beta (orange) stimulation in two example neurons, compared to the same neuron in a 6-minute no stimulation period. During the no stimulation period phase of neural spiking was determined with respect to a virtual tACS signal. The same thresholding steps to determine significance of phase precession (Supplementary Fig. S15). Like the two examples shown above, none of the 28 neurons (alpha $N = 15$, beta $N = 13$) that displayed significant phase precession during stimulation showed significant phase shifts during the no stimulation period.

b) In the human study, only 4 phases were tested, with 90 degrees difference. If I understand correctly, the precise phase was estimated by averaging data of 55 trials. I see that it is practically impossible to test a large number of different phases. But could the authors comment on the validity of this approach to show drifts of 60-70 degrees (smaller than the 90 degree difference between the phases that were tested)?

Reply: We agree that this point should be clarified in the manuscript. Importantly, our analysis of the phase data assumed von Mises distribution (a “wrapped around circle normal distribution”) of the underlying data, in the same way as common linear analyses assume Gaussian distribution

of the data. After making such an assumption, we can estimate the central location of the data, which is the circular mean of 55 trials. This can take any phase along the phase circle which corresponds to the peak of the von Mises distribution. As the 4 phase conditions we probed were equally distributed with a step of 90 degrees, we can derive any von Mises distribution that is not too narrow, because mathematically four equidistant points around the unit circle are sufficient to represent the cosine function in the equation of von Mises distribution. We added the following clarification into the Method section on page 19, lines 486-488:

“We assumed that MEP data are distributed along the phase of targeted brain oscillation approximating von Mises probability density function (a “circular normal distribution”).”

Further, we agree that we cannot exclude the possibility that the data could have deviated from the derived von Mises distribution for phase conditions in-between of our sampling. That remains an avenue for future research, as we now highlight in discussion on page 15, lines 365-368:

New discussion:

“It should be noted that using TMS only a finite number of phases can be probed (every 90 degrees in this study). Future studies with next generation closed-loop TMS systems could investigate the phase precession in more resolution by testing oscillatory phases at finer grade, such as every 30 degrees.”

c) The authors assume that there is a sort of ‘reset’ after each block, and thus data from the four blocks (for both humans and macaques) can be averaged. Is there evidence that the (plastic) state is actually reset after each break, e.g. by comparing preferred phases or other variables at the beginning of each block across the four blocks?

Reply: to address this question we investigated phase shifts per block. Phase shifts across the blocks were all in the same direction and reasonably consistent given the noisier nature of a single block analysis. In all cases, the preferred phase in the first window of a block was reset to a lower value compared to the last window of the previous block. For alpha tACS the resets were: block 2-1, $137.3^\circ - 155.4^\circ = -18.1^\circ$; block 3-2, $93.0^\circ - 148.5^\circ = -55.5^\circ$; block 4-3, $81.2^\circ - 241.6^\circ = -160.4^\circ$. For beta tACS the resets were: block 2-1, $106.5^\circ - 175.8^\circ = -69.3^\circ$; block 3-2, $104.0^\circ - 295.1^\circ = -191.1^\circ$; block 4-3, $212.0^\circ - 207.2^\circ = -4.8^\circ$.

New information was added to the supplementary data (**Supplementary Fig. S6**), and we refer to it in the results section on page 6, lines 128-129:

“Furthermore, phase shifts were consistently observed within each stimulation block (Supplementary Fig. S6)”

Supplementary Fig. S6. Phase shifts per stimulation block. Phase of the first and last window are shown. Counter-clockwise phase shifts were observed in 3 out of 4 blocks for alpha tACS, and all blocks for beta tACS. Furthermore, at the beginning of each block the phase resets: For alpha tACS the resets were: block 2-1, 137.3° - $155.4^{\circ} = -18.1^{\circ}$; block 3-2, 93.0° - $148.5^{\circ} = -55.5^{\circ}$; block 4-3, 81.2° - $241.6^{\circ} = -160.4^{\circ}$. For beta tACS the resets were: block 2-1, 106.5° - $175.8^{\circ} = -69.3^{\circ}$; block 3-2, 104.0° - $295.1^{\circ} = -191.1^{\circ}$; block 4-3, 212.0° - $207.2^{\circ} = -4.8^{\circ}$.

3) Group sizes. With 20 participants and one macaque, group sizes are a bit on the low side. I do not think that this is a serious problem here as the manuscript combines evidence from humans, animals, and a computational model, but it should be mentioned in the discussion.

Reply: We added a paragraph on limitations and suggestions for future research. Here we also mention suggestions for larger sample sizes and further exploration of brain regions. Please see page 15, lines 358-359:

“Another avenue for forthcoming studies is to investigate phase precession in larger samples, as well as in other brain regions.”

Other limitations and suggestions for future research are mentioned in the paragraph on page 15 from lines 351 to 373:

“Since the present study is the first to show ...

... further advance the understanding of network-level phase precession.”

Minor:

4) How was the ‘tACS waveform’ in the macaque study defined? Was it based on the LFP artifact or on the injected current?

Reply: Thank you for pointing out this ambiguity. The tACS waveform is based on the LFP artifact. The LFP was filtered with a 2nd order Butterworth filter between 9-11 Hz for alpha stimulation and 19-21 Hz for beta stimulation. We added this to the manuscript on page 21, lines 579-581:

“The tACS waveform was based on the LFP artifact, we quantified the PLV and neuronal phase shift using a filtered version of the LFP. The latter was filtered with a 2nd-order Butterworth filter between 9-11 Hz for alpha stimulation and 19-21 Hz for beta stimulation.”

5) In equation (7), the reader may get the impression that this is a surface integral and then get confused (at least I did). To make it easier for the reader, I would suggest calling the displacement l (instead of s , which is often used for surfaces), or directly indicating that it is a line integral.

Reply: We agree. We changed s to l in equation 7. To avoid confusion, we changed “quasipotentials (ψ)” to electric potential (V_e). To clarify, we added on page 24, lines 660-661: “Because an electric field is considered quasistatic, it can be divided into spatial and temporal components.” And refer to^{4,5}.

6) In the computational model, the authors adjust the firing rate for α - and β -tACS. It is not clear to me why the oscillation frequency of the model must be equal to the firing rate of the neurons. Couldn't the neurons keep their firing rate, and only synchronization properties are adjusted to the α - or β -range, to avoid any bias due to different firing rates?

Reply: We chose firing rates of approximately 10 and 20 spikes per second based on the assumption that these values contribute to the endogenous alpha and beta oscillations and are more susceptible to alpha and beta tACS⁶⁻⁸. However, we fully agree with the reviewer that investigating the effects at average firing rates is informative as well. Therefore, we performed further simulations of neurons with the firing rate close to the ones recorded in Experiment 2 (3.02 spikes per second at baseline in the alpha condition, 2.74 spikes per second baseline in the beta condition). The results are similar to the original analysis with stronger entrainment in the anterior and posterior wall of the precentral gyrus, compared to the crown. Entrainment of the anterior wall (BA6) is more likely at phases between 90 and 180°, whereas entrainment in the posterior wall (BA4) occurs at phases between -90° and 0°. We added this analysis to the supplementary materials (**Supplementary Fig. 25**) and refer to this result on page 11, lines 237-239:

“We repeated the analyses with intrinsic firing rates as observed in the baseline measurements of experiment 2 (3.02 spikes per second and 2.74 spikes per second for the alpha and beta blocks respectively). The results of these analyses were similar to the findings mentioned above (Supplementary Fig. S25).”

Supplementary Fig. 25. Simulations equivalent to those in Fig. 4, but with intrinsic firing rates based on baseline recordings of experiment 2 (3.02 spikes per second and 2.74 spikes per second for the alpha and beta blocks respectively). The results are similar to the original analysis with (A) stronger entrainment in the anterior and posterior wall of the precentral gyrus, compared to the crown. B) Entrainment of the anterior wall (BA6) is more likely at phases between 90 and 180°, whereas entrainment in the posterior wall (BA4) occurs at phases between -90° and 0°.

Bettina Schwab

Reviewer #2 (Remarks to the Author):

In this manuscript, Wischnewski, Tran, and their colleagues aim to establish a causal link between LFP dynamics and phase precession, as stated in the abstract. The authors employ a unique approach, combining human and monkey data with computational, multi-scale models.

I have three core comments that I believe need to be addressed in order to improve/strengthen the study:

A) In my opinion, the connection between the study's findings and physiological phase precession is not clear. I personally do not think that there is enough evidence to make specific inferences about physiological processes, considering that the phase precession discussed in the manuscript is relative to exogenous stimulation. At least, the authors should provide convincing reasons, for instance in the discussion, of why they think that the exogenous manipulations they applied may provide direct insights about physiological phase precession.

B) To enhance the robustness of the results, further analysis is necessary, particularly considering that several arbitrary thresholds are applied to the data for Experiment 2. A more thorough examination of those thresholds and their impact on the analysis outcomes would strengthen the results of this manuscript.

C) The computational model in the study should be utilized more extensively to elucidate the underlying mechanisms behind the reported effects. By using the model in a more comprehensive manner, the authors can provide a deeper understanding of the observed effects of stimulation on neuronal activity.

Below, you can find specific comments related to each of the previous points.

Reply: We are sincerely grateful to the reviewer's contributions and insightful feedback on this study. We believe that by addressing these thoughtful suggestions the quality of the manuscript has significantly improved.

To summarize, we provide additional analyses to show:

- 1) Results in human data are not driven by outliers.
- 2) There is no association between individual phase shifts and increase in cortical excitability.
- 3) There is no association between arousal and changes in excitability or phase.
- 4) Results are robust for different thresholding criteria.
- 5) Phase shifts are unlikely to reflect false positives.
- 6) Population modeling results with average intrinsic firing rates.
- 7) Results of alternative microcircuit models.
- 8) Contributions of AMPA and GABA plasticity in the original microcircuit model.

Please see a detailed response to all comments below.

A.

1 - It appears that the paper's main focus is on the effects of exogenous electrical stimulation on brain activity, rather than directly on physiological phase precession. In particular, there is a lack of LFP recordings, which diminishes the direct association between LFP dynamics and the study's findings. In my opinion, the paper would benefit from emphasizing the effects of AC stimulation instead of attempting to establish connections to physiological processes.

Reply: One of the mechanisms, which in this work we assumed to reflect phase precession, is that neurons adapt to a modulation of the extracellular electric field. It is a coordination of neural spiking timing to changes in the LFP⁹. In previous studies the modulation of LFP has been induced by specific task or environment. In our study, the modulation of LFP occurs directly through external application of alternating currents¹⁰⁻¹². We believe that the means through which LFP modulation occurs (task-driven or external) is inconsequential for cellular-level phase precession. Neural firing adapts to changes in extracellular electric fields, regardless of how these changes occur. As such, we are of the opinion that the AC stimulation directly alters cellular physiological processes and that our study results warrant a discussion on phase precession.

2 - On a broader note, phase precession is typically believed to facilitate coding, learning, or plasticity (as also stated by the authors). However, since the subjects in this study are not actively engaged in any specific tasks or behaviors, it raises the question of what would be the purpose of phase precession in this situation. At least, further exploration and discussion of the potential functional significance in the absence of specific tasks would add depth to the study.

Reply: Thank you, this is a great point. Previous studies have investigated phase precession in the context of a task, where it naturally occurs. In this study, we were primarily interested in the mechanistic understanding of phase precession. Therefore, we opted for an approach in which LFPs are directly modulated in a relatively confined brain area (by an external AC field), rather than indirectly in the whole brain (through a task). Behavioral consequences are therefore difficult to assess in the present study, as such we can only speculate. We provide a short discussion on page 15, lines 351-356:

“Since the present study is the first to show AC stimulation-induced phase precession, follow up questions arise that may inspire future research. Our study focused on the mechanistic understanding of phase precession, yet it would be fascinating to study the functional consequences of the observed effects. One hypothesis is the signaling between premotor and primary motor regions becomes more efficient. Premotor-to-motor connectivity is crucial for motor learning^{13,14} and is abnormal in disorders of motor control^{15,16}. Furthermore, tACS to motor regions has shown to facilitate motor learning¹⁷⁻¹⁹.”

3 - Is it possible that the observed phase precession is merely a result of LFP entrainment? If a neuron maintains a fixed spiking phase relative to the LFP, and the LFP becomes progressively entrained by the stimulation, it could lead to an apparent shift in the neuron's phase relative to the stimulation. However, in this case, such a shift would not correspond to actual phase precession (the phase would be unchanged relative to the LFP). Are there any obvious reasons to exclude this hypothesis?

Reply: This is an intriguing question. Previous basic studies found that LFP entrainment through application of AC stimulation happens instantaneously^{20,21}. That is, an intrinsic LFP and AC-induced electric field, both being an electromagnetic field from physics standpoint, immediately superimpose (merge together) and act as one. However, we observe gradual shifts in neural spiking phase preferences over the course of 6 minutes, which suggests an adaptation of neural firing in the form of phase precession, rather than adaptation of LFP, as the most plausible explanation for our observations.

4 - The phase changes depicted in Fig. 2 appear rather abrupt and may not align with the typical definition of phase precession. Can they be called “phase precession”?

Reply: We agree with the reviewer that the observed phase shifts in the human data appear less gradual than in the single-neuron data. Motor-evoked potential (MEP) measurements represent a system level response by a population of central and peripheral nervous system neurons. Thus, MEPs depend on the immediate brain and muscle states (much more than a single-neuron activity does) and naturally experience considerable trial-to-trial variability²²⁻²⁴. Given that, we do believe that the observed phase shifts are relatively continuous, and the observed phase shifts are reminiscent of phase precession. In agreement with the reviewer suggestion, we decided to reserve the terminology “phase precession” for experiments 2 and 3. For phase changes experiment 1 we now refer to “phase shift”. Furthermore, in the discussion section we elaborate on the translatability of neural phase precession to network-level phase shifts. Please see page 15, lines 368-373:

“Finally, the translation from single cell phase precession to network-level phase shifts may be explored in further detail. One way would be via combining intracortical and non-invasive recordings. Another possibility comes with the advancement of multiscale computational modeling. TMS, which was used to probe cortical excitability in this study, generates a complex cascade of direct- and indirect-waves²⁵. As such, integrating modeling of AC fields and TMS dynamics would further enhance the understanding network-level phase precession.”

B.

5 - In Fig. 2, it is generally unclear whether the reported results are aggregated across subjects. Are the results consistent across subjects or are they driven by a few?

Reply: This is correct, the results of Fig. 2 represent the averaged results across subjects. We clarified this in the figure legend.

To identify whether the results are driven by outliers, we performed an N-2 sub-sample analysis equivalent to 10-fold cross-validation. Specifically, we investigated the preferred phase for 10 different sub-samples of N=18 (i.e., results of 2 participants were left out every repetition). As shown below, the subsamples yield comparable results, suggesting that results are not distorted by outliers. We included these results in the supplementary materials (**Supplementary Fig. 7**) and refer to these results on page 6, lines 129-130:

“Resampling with N-2 subgroups suggested that the observed phase shifts were not driven by outliers (Supplementary Fig. S7).”

Supplementary Fig. S7. Results of an N-2 sub-sampling analysis. This analysis was performed to exclude the possibility that the results of experiment 1 were a consequence of outliers. Specifically, polar plots were generated for 10 sub-samples of $n = 18$ (leaving out $n = 2$). Each dot represents one sub-sample. For clarity, we show three of the 20 windows: window 1, 11 and 20. Results were consistent for all sub-samples, for both alpha and beta stimulation, suggesting that the reported main results are not driven by outliers.

6 - Could the general increase in MEP size (Supplementary Fig. S5) potentially confound the reported results? For instance, is there any correlation between the phase shift and the slope of the MEP size increase across subjects?

Reply: As the order of phase-targeting with TMS is randomized within each block, we anticipated no correlation between the cortical excitability increase and observed phase shift. To verify this, we correlated for each participant the window-to-window increase in cortical excitability with the shift in phase. We found no consistent relationship between both measures. On average, the correlation across participants was $R = -0.067$ ($p = 0.29$) and $R = -0.007$ ($p = 0.91$) for alpha and beta tACS respectively. We added these findings to the supplementary materials (**Supplementary Fig. S10**) and refer to these on page 7, lines 134-135:

Supplementary Fig. S10. Individual correlations between change in cortical excitability and phase shifts. For each participant ($n = 20$) the window-to-window change of both measures were correlated. On average, $r = -0.067$ ($p = 0.29$) for alpha stimulation and $r = -0.007$ ($p = 0.91$) for beta stimulation. This suggests that the observed phase shifts (main Fig. 2) and changes in cortical excitability (Supplementary Fig. 8, 9) are independent.

*“These increases in excitability were independent of phase shifts (**Supplementary Fig. S10**)”*

7 - Is there a correlation between the MEP size and changes in the subjects' state? Is there, for instance, any systematic change in pupil size, as a proxy for arousal, which may explain the MEP size drift and/or maybe phase changes?

Reply: In the human experiment, no pupil size data was collected. However, we systematically collected questionnaire data on visual and somatic sensations during tACS, and how tiring tACS was perceived. We calculated the Spearman rank correlation between each item and individual MEP amplitude change (window 20 – window 1), as well as phase shift (window 20 – window 1). We found no effect association between changes in subject state and cortical excitability increase or phase shifts ($p > 0.3$). Therefore, we believe that changes in subject arousal are an unlikely explanation for the observed effects on MEP amplitude and phase. These results were added to the supplementary materials (Supplementary Fig. S11), and we refer to these results on page 7 lines 135-137:

“Furthermore, the observed changes in phase preference and excitability are not associated with subjective measures on participants’ arousal (Supplementary Fig. S11).”

Supplementary Fig. S11. Spearman-rank correlations between changes in subjective subject states and cortical excitability changes and phase shifts. Neither visual sensations (phosphenes), nor somatic sensations (skin tingling or itching), nor how tiring the stimulation was perceived was significantly correlated to individual changes in cortical excitability or phase shifts. This suggests that the main results are not confounded by changes in participants’ arousal states.

8 - Is Experiment 2 based on a single recording session?

Reply: That is right, the experiments on the non-human primate data (experiment 2) were collected in a single session. First, four blocks of alpha tACS followed by four blocks of beta tACS. Time between alpha and beta tACS was approximately 25 minutes. Time between blocks was approximately 3 minutes. Please see page 20, lines 522-524:

“For each condition, we have four stimulation blocks of 6 minutes each separated by a 3-minute period representing a total of 8 blocks in a single session.”

9 - The methods section lacks information regarding the definition of PLV in the baseline. How is it computed since there is no stimulation in that period?

Reply: Thank you for highlighting this missing information. We added the following details about the PLV computation in the manuscript on page 21, lines 581-583:

“For the no stimulation period we computed the PLV and neuronal phase shift with regards to a virtual tACS waveform with the corresponding block frequency (10Hz or 20Hz, for alpha and beta stimulation respectively).”

10 - In line 489, it is mentioned that various metrics were utilized to assess the stability of the units; however, there is no explicit mention or reporting of the specific analyses conducted. If plots or additional information demonstrating the stability of the recordings and excluding recording drift are available, it would be beneficial to include them in the manuscript. Examining such plots or analyses would help validate the reliability of the results and provide a clearer understanding of whether the observed phase changes are influenced by recording drift or other factors.

Reply: We fully agree that the addition of this information is important. We added a supplementary figure (**S14**) in the manuscript showing 4 single unit waveforms and their corresponding inter-spike interval (ISI), as well as the mean spike amplitude and standard deviation. Additionally, we added a histogram of the signal-to-noise ratio (SNR) of the 81 units recorded. We refer to the new figure on page 21, lines 566-567:

*“Overall SNR, as well as examples of extracellular spike waveforms are shown in **Supplementary Fig. 14.**”*

Supplementary Fig. S14. Examples of cluster of single units recorded. A) Extracellular spike waveforms of 4 isolated units are shown with additional features. The ISI of the cell is shown in the bottom right with the total number of spikes detected during the recording period. In the top left corner is the amplitude of the spike (mean \pm standard deviation, unit: μV). The thick black line represents the mean average waveform. The black line in the bottom left of each graph represents 200 μs . B) Histogram of the signal-to-noise ratio (SNR) of the cells recorded (3.50 \pm 0.73, mean \pm standard deviation).

11 - Understanding the robustness of the phase change results in Experiment 2 is challenging due to the extensive pre-processing described in the Methods section (lines 511-518). It is crucial to determine if the observed phase shift, which is the main result derived from these experiments, remains robust when altering the thresholding parameters. How do the results change when these thresholds are modified? Additionally, I am uncertain about the application of the final threshold ($> 15^\circ$). Is the data analyzed for phase shifts limited to neurons that already exhibited a phase shift? If this is the case, it might be expected to find neurons with phase shifts. Could the authors clarify this point?

Reply: In order to clarify the process, we added a supplementary figure (S15) explaining the different thresholds and modify the text accordingly. We include this now in the results section on page 8, lines 174-178:

“To classify neuron behaviors based on their phase shifts, we created a framework that could be divided into three steps / thresholding: 1) we keep neurons that are responsive in at least 10 time windows (or 50% of the total number of time windows), 2) we select neurons that exhibit a significant circular-linear correlation and finally 3) we keep neurons exhibiting a phase shift greater than 15° between the preferred phase in the first time window and the last time window.”

Supplementary Fig. S15. Schematic of the framework used to quantify neurons behaviors. Once all the neurons were detected, we apply a 3 thresholding steps: First, we only kept neurons that are responsive in at least 10 time windows – corresponding to 50% of the total number of time windows ($N=20$) using a Rayleigh uniformity test. Second, neurons were only kept if they exhibited a significant circular-linear correlation ($p < 0.05$). Third, we kept neurons exhibiting a phase shift (Φ) greater than 15° in either direction.

The data analysis was limited to neurons that meet the three thresholds: they must be responsive in at least 10 time windows (out of 20), exhibit a significant linear-circular correlation ($p < 0.05$), and this phase shift should be of practically meaningful extend (greater than 15°). We introduced these thresholds to avoid putative situation when the phase shift is statistically significant but practically meaningless (of a too small effect). To assess the robustness of thresholding we further explored 4 different effect size thresholds - minimum phase shifts of 10° , 20° , 25° , and 30° . Higher thresholds lead to more conservative categorization. However, the general pattern of results and conclusions we can draw remain unaffected by a specific threshold. New results were added to the supplementary data (Supplementary Table 1), and we refer to these results on page 9, line 197: “Results for different angle thresholds are shown in Supplementary Table 1”.

Supplementary Table 1. Influence of the shift value on the number of neurons. The default shift value we used in the main manuscript is 15° . The higher the threshold value is, the more selective we are. Even with very high value, few neurons exhibit a positive/negative phase shift.

Alpha stimulation

Shift (°)	10	15	20	25	30
Positive	10	7	3	3	2
Negative	10	8	8	7	5
Total	20	15	11	10	7

Beta stimulation

Shift (°)	10	15	20	25	30
Positive	14	9	4	3	2
Negative	7	4	3	3	3
Total	21	13	7	6	5

12 - Another related question arises regarding the application of a threshold on phase shifts. The data presented, such as in Figure 2B, appears to have gaps in the middle of the distribution due, I guess, to the applied threshold. If these missing data points were included, one might expect the entire distribution to appear more continuous. Hence, it raises the question of whether the observed phase shifts are genuine or merely a result of fluctuations. Is it possible to simulate a ground truth to shed light on this matter? For instance, by simulating a population of neurons with similar firing rate distributions as in the experimental data and for a similar amount of time, the authors could investigate what the distribution of phase shifts would look like and quantify the percentage of false-positives.

Reply: Due to the different nature of data, the way that phase (and thus shifts) is calculated in the human experiment (experiment 1, figure 2) differs from the NHP experiment (experiment 2, figure 3). For the human experiment MEPs were collected at four phases of tACS (0°, 90°, 180°, 270°). At each trial the TMS was applied at one of these phases and the order of phases was random. During analysis, the polar vector of each data window (55 trials) was calculated, which results in an estimated preferred phase per time window. Note that for this analysis we included all data, and no thresholding was applied, nor were any data points removed. As we argued in point 4, apparent discontinuity is likely a result of variability within the MEP measurements.

Supplementary Fig. S5. Permutation testing on circular-linear correlations in experiment 1. Phase information of MEP was shuffled in 10000 permutations. Actual observed correlation values for alpha ($r = 0.655$) and beta ($r = 0.825$) both fall in the 95th percentile of the permutation distribution ($p = 0.041$ and $p > 0.001$, respectively), suggesting that the observed phase shifts in human data are unlikely to occur spuriously.

To determine the likelihood that the observed phase shifts in humans reflect a false-positive, we performed a permutation test with 10000 permutations with MEPs at randomized phases. The results of this analysis suggest that the actual observed correlations (alpha $r = 0.655$, beta $r = 0.825$) are unlikely to be a result of false positives (alpha $p = 0.041$, beta $p < 0.001$). We added this analysis to the supplementary materials (**Supplementary Fig. 5**) and refer to it in the main text on **page 6, lines 127-128**:

“Permutation testing on excitability with randomized phases suggested that spurious uniform phase shifts in our data are unlikely ($p < 0.05$, Supplementary Fig. S5).”

13 - Could the authors clarify the definition of a 'non-uniform' phase shift for neurons? What are the specific quantitative differences between neurons with non-uniform phase shifts and those exhibiting a 'uniform' phase shift? Additionally, is the finite duration of the experiment (relative to the firing rate distributions) a potential concern for this classification? Could it be possible, for instance, that neurons with phase shifts are defined as such just because they do not fire much and therefore it is easier to find periods where it seems like there is a uniform shift? Similarly, is it possible that some units with non-uniform phase shifts would simply experience their phase change at a later time? I also noticed that some neurons in Figures S7-8 exhibit similar time profiles to the examples shown in Figures S10-11. Could the authors provide further insight into this observation?

Reply: Neurons that we categorized as having a uniform phase shift were those that remained after the three thresholding steps. Please see question 11 above (and now added as supplementary figure 15). That is, these neurons were 1) responsive in ≥ 10 windows, 2) had a significant circular-linear correlation, and 3) displayed a phase shift of larger than $|15^\circ|$. The neurons that we classified as 'non-uniform' (Supplementary Fig. S19B and S20B) were responsive in ≥ 10 windows, but did not display a significant circular-linear correlation (threshold step 2). Thus, although entrained, their phase movement did not follow a unidirectional or uniform path. We have not further analyzed 'non-uniform' neurons, because capturing all possible nonlinear dynamics requires a specific study design and is outside of our original question of interest.

So even if superficially some neurons in Supplementary Figure S19 and S20 (before S10 and S11) resemble those in S16 and S17 (before S7 and S8), they are statistically not the same and the latter follow a linear path, whereas the former do not. We clarified this definition in the legends of **supplementary figures S19 and S20**:

“These were defined as neurons that were responsive in ≥ 10 windows but did have no significant circular-linear correlation.”

Further, we found no evidence for an association between the absolute phase shift (in degrees) and firing rate (in Hz) – Spearman rank correlation, $r = -0.23$, $p = 0.22$.

It is certainly possible that longer stimulation durations may affect phase shift trajectories beyond what we observed here. Future studies may explore the effects of longer stimulation periods. We cover this point in the added section on future directions (**page 15, lines 356-358**):

“While we observed phase precession in stimulation periods of approximately 6 minutes, it is possible that some neurons respond slower. Therefore, stimulation durations should be systematically investigated.”

14 - As for Experiment 1, are there any signs that the animal's arousal state is stable during the stimulation period in Experiment 2?

Reply: The animal was seated alone in a recording chamber without performing any type of task and the experimenter did not enter the room to avoid disturbance. The animal was constantly monitored via video camera to ensure no signs of distress during the whole recording session. Please see page 20, lines: 533-534:

“During the whole session, the animal was remotely monitored to ensure no signs of distress.”

C.

15 - In Experiment 3a the firing rate was set at 10 and 20 spikes per second. Why was the firing rate set like that? It would seem that a more reasonable choice would be to fix the rates to the ones recorded in Experiment 2.

Reply: We chose firing rates of approximately 10 and 20 spikes per second based on the assumption that these values contribute to the endogenous alpha and beta oscillations and are more susceptible to alpha and beta tACS⁶⁻⁸. However, we fully agree with the reviewer that investigating the effects at average firing rates is informative as well. Therefore, we performed further simulations of neurons with the firing rate close to the ones recorded in Experiment 2 (3.02 spikes per second at baseline alpha, 2.74 spikes per second baseline beta). The results are similar to the original analysis with stronger entrainment in the anterior and posterior wall of the precentral gyrus, compared to the crown. Entrainment of the anterior wall (BA6) is more likely at phases between 90 and 180°, whereas entrainment in the posterior wall (BA4) occurs at phases between -90° and 0°. We added this analysis to the supplementary materials (**Supplementary Fig. 25**) and refer to this result on page 11, lines 237-239:

“We repeated the analyses with intrinsic firing rates as observed in the baseline measurements of experiment 2 (3.02 spikes per second and 2.74 spikes per second for the alpha and beta blocks respectively). The results of these analyses were similar to the findings mentioned above (Supplementary Fig. S25).”

Supplementary Fig. 25. Simulations equivalent to those in Fig. 4, but with intrinsic firing rates based on baseline recordings of experiment 2 (3.02 spikes per second and 2.74 spikes per second for the alpha and beta blocks respectively). The results are similar to the original analysis with (A) stronger entrainment in the anterior and posterior wall of the precentral gyrus, compared to the crown. B) Entrainment of the anterior wall (BA6) is more likely at phases between 90 and 180°, whereas entrainment in the posterior wall (BA4) occurs at phases between -90° and 0°.

16 - What could be the reason for observing phase precession that is independent of the LFP frequency, as indicated by the generic phase shift observed in many units for both alpha and beta stimulation in Figure S9? Why are the phases and phase shifts different for the two stimulation frequencies? Could the model be used to understand this observation?

Reply: These are great questions. Given that we use a simplified model we cannot provide a full mechanistic explanation of all possible phase shifts as observed in experiment 2. The model used in the main results – containing (I) one BA6 pyramidal neuron, (II) one BA4 pyramidal neuron, and (III) one BA4 inhibitory interneuron – is the simplest model we could think of that explains the results observed in experiment 1. To further illustrate that, we now added two alternative simplified networks. First, a model (A) with a pyramidal-pyramidal connection at opposing neuron orientations and without inhibitory interneuron. This would reflect an uninhibited connection from BA6 to BA4. In contrast to the original model, this model showed a clockwise phase shift with increasing synaptic weights. Thus the inhibitory neurons likely contribute to resulting direction of the phase shift. The second new model B contained a pyramidal-pyramidal connection at the same neuron orientations and without inhibitory interneuron. This would reflect an uninhibited connection within BA4. The results of this model showed no prominent phase shift. The results of these alternative models was added to the supplementary materials (**Supplementary Fig. 26**). Furthermore, we refer to these results on page 11, line 246:

“Two alternative models are explored in **Supplementary Fig. 26**”

Supplementary Fig. S26. We explored the effects of two alternative simplified network models. A) a model with a pyramidal-pyramidal connection with opposing neuron orientations and without inhibitory interneuron. This would reflect an uninhibited connection from BA6 to BA4. In contrast to the original model in the main results, this model showed a clockwise phase shift with increasing synaptic weights, going from $\sim 240^\circ$ to $\sim 180^\circ$ for alpha stimulation, and going from $\sim 280^\circ$ to $\sim 230^\circ$ for beta stimulation. B) A model with a pyramidal-pyramidal connection with the same neuron orientations and without inhibitory interneuron. This would reflect an uninhibited connection within BA4. The results of this model showed no prominent phase shift for either stimulation frequencies (Fig. 4E).

Comparing all models, the originally used model is the best explanation for phase shifts observed in the human motor cortex. The alternative models show that the nature of the phase precession (angles and direction) strongly depends on specific network connections. To some extent this explains the differences in phase shifts between different neurons, as observed in experiment 2. We discuss this on page 14, lines 336-338:

“In two alternative models, which did not contain GABAergic connections, phase precession direction and trajectory were significantly altered (Supplementary Fig. S26). This hints towards phase precession being network-dependent, which is in line with the various phase shift trajectories observed in different neurons of the NHP data.”

Nevertheless, an exhaustive computational understanding of phase precession mechanisms (including effects of different frequencies) requires systematic testing of more complex network models, which include various neuron types and take into account layer-specific connections. We believe that such an investigation is fascinating for future research but is beyond the scope of the present study. We mention this limitation on page 14, lines 339-341:

“An exhaustive understanding of network dynamics on phase precession requires systematic testing of complex models with a variety of neuron types and layer-specific connections, which is beyond the scope of the present study.”

17 - While the clarity of the results in Fig. 4A-C is evident, I think that this large scale simulation could be more valuable if the authors were estimating the relative contribution of multiple factors, such as electric field magnitude, direction, neuronal somatodendritic axis and specific cell examples (5), to the estimated PLV.

Reply: The ROI of the motor cortex in the model of experiment 3A is populated with five clones of neurons at different orientations and consequently receive different magnitude of electric fields. We agree with the reviewer that it would be interesting to explore this further by using a large scale simulation and applying different stimulation montages and different intensities. However, the goal of experiment 3A was to provide simulations with regards to the human data (experiment 1). As such, elaborate systematic modeling of different scales and stimulation setups, although interesting, falls outside the scope of this study. We mention such exploration as a future direction on page 15, lines 359-365:

“The prefrontal cortex is associated with various cognitive functions and abnormal plasticity in this region relates to psychiatric disorders²⁶. As such, investigating AC-induced phase precession in the prefrontal cortex, by for example investigating TMS-evoked potentials, can be of clinical importance²⁷. In tandem with exploring effects in other brain areas, computational modeling of electric fields could be used to investigate different stimulation montages. Thereby, the effects of AC stimulation on different neural orientations and cell types can be studied. Additionally, systematically exploring various intensities would allow for establishing a dose-response curve for externally induced phase precession²⁸.”

18 - Why are there neurons in the model for which the PLV decreases relative to baseline?

Reply: This is an intriguing question. Decreased PLV compared to baseline has been observed previously, particularly in neurons that receive very small electric fields¹. As such, we hypothesized that the neurons with decreasing PLV would be those that receive the smallest electric field. Figure (A) below shows the amplitude of the component of electric field normal to surfaces. Neurons in the gyral crown receive the smallest electric fields (normal component), whereas neurons in the gyral walls receive the highest electric fields (normal component). Figure (B) below shows the location where PLV of L5 pyramidal neurons is lower compared to the baseline condition. As anticipated, these regions match the area with very low normal component of electric field. Further some neurons with such behavior are located at the edge (boundary) of the model and may be affected by the boundary inaccuracies inherent for numeric simulations.

19 - I think that the model should be explored more to answer a specific set of questions. For instance, if the inputs are boosted, something that would provide an increased output that would mimic increased MEP size, does the phase change?

Reply: We agree that it would be fascinating to have a model that integrates both electric fields of tACS and stimulation of TMS. However, single pulse TMS generates a rather complex cascade of responses, including direct and indirect waves (D-waves and I-waves) in various neuronal pools of the motor cortex²³. Current models that have attempted to model TMS responses (e.g. ²⁵) have not yet yielded good predictive power. Simply integrating a TMS response in our current model is therefore not trivial. Expanding our model to include TMS responses would be an entire study by itself and is beyond the scope of the present study. We further discuss the implementation of integrated AC-TMS modeling approaches on page 15, lines 368-373:

“Finally, the translation from single cell phase precession to network-level phase shifts may be explored in further detail. One way would be via combining intracortical and non-invasive recordings. Another possibility comes with the advancement of multiscale computational modeling. TMS, which was used to probe cortical excitability in this study, generates a complex cascade of direct- and indirect-waves²⁵. As such, integrating modeling of AC fields and TMS dynamics would further enhance the understanding network-level phase precession.”

20 - While the hypothesis of NMDA plasticity is reasonable, it is worth considering that there might be multiple parameters that could potentially induce phase shifts. Exploring alternative possibilities could be beneficial if the authors aim to determine the primary candidate responsible for the observed phase shifts.

Reply: We thank the reviewer for the suggestions. We explored two alternative candidates that may be responsible for the phase shifts by systematically changing the BA6-PY to BA4-PY AMPA or GABA_A synaptic weights. For AMPA, we utilized the same synaptic weight step as NMDA. We observed phase shifts that were similar to NMDA, although it was less clear for beta stimulation. Initially, for GABA_A we did not observe any phase shifts using the same synaptic weights steps as for NMDA. To ensure that GABA_A is not associated with phase shifts we multiplied synaptic weights steps for GABA_A by 10. Still no phase shifts were observed, suggesting that our experimental data cannot be explained by synaptic changes in GABA_A. We added these additional models to the supplementary materials (**Supplementary Fig. 27**). We refer to these results on page 12, lines 252-258:

“Additionally, we found similar phase shifts when increasing synaptic strength of α -amino-3-hydroxy-5-methyl-4-isoxazolepropionic acid (AMPA) for alpha stimulation (Supplementary Fig. 27A). For beta stimulation AMPA strength-related phase shifts were slightly weaker compared to NMDA. Increasing strength of γ -Aminobutyric acid-A (GABA_A) synaptic connections did not cause any phase shifts (Supplementary Fig. 27B). Together, this suggests that AMPA- and NMDA-mediated synaptic plasticity, but not GABA_A-mediated plasticity, between premotor and primary motor neurons captures the observed shifts in preferred phase of cortical motor output.”

Supplementary Fig. S27. Within the original network model (One PY in BA6, one PY in BA4, and one IN in BA4, Fig. 4D) we investigated the contribution of synaptic weight changes of AMPA and GABA_A connections to the BA4 PY neuron. A) For AMPA, we used the same synaptic weight step as NMDA. We observed phase shifts that were similar to NMDA, particularly for alpha stimulation (shift from ~90° to ~150°). For beta stimulation also a counter-clockwise phase shift was observed (from ~110° to ~150°), but the trajectory fitted the experimental data less well compared synaptic changes after NMDA. B) For GABA_A we did not observe any phase shifts, even if synaptic weights steps were multiplied by 10 compared to AMPA and NMDA. To ensure that GABA_A is not associated with phase shifts we multiplied synaptic weights steps for GABA_A by 10. This suggests that our experimental data cannot be explained by synaptic changes in GABA_A.

Minor comments

21 - The concluding sentence in the discussion, in its current form, seems better suited for an introduction or abstract rather than as the final sentence of the paper.

Reply: We agree, and added the following sentence (page 15, lines 383-384):

“Altogether, the present study demonstrated that the shifting of preferred phase is one key mechanism by which tACS modulates neural activity.”

22 - Line 570: typo “electric fields on neuros”

Reply: we corrected the typo.

23 - Incorrect reference in Supplementary Fig. S6. “Similar data as in Supplementary Fig. 2”. I think it should be Supplementary Fig. 5.

Reply: This was corrected.

References:

1. Tran, H., Shirinpour, S. & Opitz, A. Effects of transcranial alternating current stimulation on spiking activity in computational models of single neocortical neurons. *NeuroImage* **250**, 118953 (2022).
2. Reato, D., Bikson, M. & Parra, L. C. Lasting modulation of in vitro oscillatory activity with weak direct current stimulation. *J. Neurophysiol.* **113**, 1334–1341 (2015).
3. Esmailpour, Z., Kronberg, G., Reato, D., Parra, L. C. & Bikson, M. Temporal interference stimulation targets deep brain regions by modulating neural oscillations. *Brain Stimulat.* **14**, 55–65 (2021).
4. Plonsey, R. & Heppner, D. B. Considerations of quasi-stationarity in electrophysiological systems. *Bull. Math. Biophys.* **29**, 657–664 (1967).
5. Bossetti, C. A., Birdno, M. J. & Grill, W. M. Analysis of the quasi-static approximation for calculating potentials generated by neural stimulation. *J. Neural Eng.* **5**, 44–53 (2008).
6. Fröhlich, F. & Riddle, J. Conducting double-blind placebo-controlled clinical trials of transcranial alternating current stimulation (tACS). *Transl. Psychiatry* **11**, 1–12 (2021).
7. Huang, W. A. *et al.* Transcranial alternating current stimulation entrains alpha oscillations by preferential phase synchronization of fast-spiking cortical neurons to stimulation waveform. *Nat. Commun.* **12**, 3151 (2021).
8. Zhao, Z., Shirinpour, S., Tran, H., Wischnewski, M. & Opitz, A. Intensity- and frequency-specific effects of transcranial alternating current stimulation are explained by network dynamics. 2023.05.19.541493 Preprint at <https://doi.org/10.1101/2023.05.19.541493> (2023).
9. Qasim, S. E., Fried, I. & Jacobs, J. Phase precession in the human hippocampus and entorhinal cortex. *Cell* **184**, 3242–3255.e10 (2021).
10. Anastassiou, C. A. & Koch, C. Ephaptic coupling to endogenous electric field activity: why bother? *Curr. Opin. Neurobiol.* **31**, 95–103 (2015).
11. Anastassiou, C. A., Perin, R., Markram, H. & Koch, C. Ephaptic coupling of cortical neurons. *Nat. Neurosci.* **14**, 217–223 (2011).
12. Fröhlich, F. & McCormick, D. A. Endogenous electric fields may guide neocortical network activity. *Neuron* **67**, 129–143 (2010).
13. Nitsche, M. A. *et al.* Contribution of the Premotor Cortex to Consolidation of Motor Sequence Learning in Humans During Sleep. *J. Neurophysiol.* **104**, 2603–2614 (2010).
14. Kantak, S. S., Stinear, J. W., Buch, E. R. & Cohen, L. G. Rewiring the Brain: Potential Role of the Premotor Cortex in Motor Control, Learning, and Recovery of Function Following Brain Injury. *Neurorehabil. Neural Repair* **26**, 282–292 (2012).
15. Weissbach, A. *et al.* Premotor–motor excitability is altered in dopa-responsive dystonia. *Mov. Disord.* **30**, 1705–1709 (2015).
16. Suppa, A. *et al.* Dopamine Influences Primary Motor Cortex Plasticity and Dorsal Premotor-to-Motor Connectivity in Parkinson’s Disease. *Cereb. Cortex* **20**, 2224–2233 (2010).
17. Pollok, B., Boysen, A.-C. & Krause, V. The effect of transcranial alternating current stimulation (tACS) at alpha and beta frequency on motor learning. *Behav. Brain Res.* **293**, 234–240 (2015).

18. Krause, V. *et al.* Cortico-muscular coupling and motor performance are modulated by 20 Hz transcranial alternating current stimulation (tACS) in Parkinson's disease. *Front. Hum. Neurosci.* **7**, (2014).
19. Krause, V., Meier, A., Dinkelbach, L. & Pollok, B. Beta Band Transcranial Alternating (tACS) and Direct Current Stimulation (tDCS) Applied After Initial Learning Facilitate Retrieval of a Motor Sequence. *Front. Behav. Neurosci.* **10**, (2016).
20. Krause, M. R., Vieira, P. G., Csorba, B. A., Pilly, P. K. & Pack, C. C. Transcranial alternating current stimulation entrains single-neuron activity in the primate brain. *Proc. Natl. Acad. Sci. U. S. A.* **116**, 5747–5755 (2019).
21. Johnson, L. *et al.* Dose-dependent effects of transcranial alternating current stimulation on spike timing in awake nonhuman primates. *Sci. Adv.* **6**, eaaz2747 (2020).
22. Groppa, S. *et al.* A practical guide to diagnostic transcranial magnetic stimulation: Report of an IFCN committee. *Clin. Neurophysiol.* **123**, 858–882 (2012).
23. Siebner, H. R. *et al.* Transcranial magnetic stimulation of the brain: What is stimulated? – A consensus and critical position paper. *Clin. Neurophysiol.* **140**, 59–97 (2022).
24. Goldworthy, M. R., Hordacre, B. & Ridding, M. C. Minimum number of trials required for within- and between-session reliability of TMS measures of corticospinal excitability. *Neuroscience* **320**, 205–209 (2016).
25. Wilson, M. T., Moezzi, B. & Rogasch, N. C. Modeling motor-evoked potentials from neural field simulations of transcranial magnetic stimulation. *Clin. Neurophysiol.* **132**, 412–428 (2021).
26. Goto, Y., Yang, C. R. & Otani, S. Functional and Dysfunctional Synaptic Plasticity in Prefrontal Cortex: Roles in Psychiatric Disorders. *Biol. Psychiatry* **67**, 199–207 (2010).
27. Tremblay, S. *et al.* Clinical utility and prospective of TMS–EEG. *Clin. Neurophysiol.* **130**, 802–844 (2019).
28. Alekseichuk, I., Wischniewski, M. & Opitz, A. A minimum effective dose for (transcranial) alternating current stimulation. *Brain Stimulat.* **15**, 1221–1222 (2022).

REVIEWER COMMENTS

Reviewer #1 (Remarks to the Author):

The authors provided additional descriptions, simulations and analyses that relieve my major concerns, and I agree with the authors that they have significantly improved their manuscript. I have two minor/clarification points left [related to previous major points]:

a) 'Gradual' shifts in spike timing / phase precession.

The authors argue with different strategies [Major 2a] for human data (shuffling of phase information) and animal data (comparison to sham condition). Could the authors explain this choice, or should one assume that the respective other strategy would not have been successful?

Similarly, the 'resets' [Major 2c] of phase shifts were discussed for the human data, but not for the animal data. Were the phase shifts similarly consistent across stimulation blocks in the animal data?

b) Suppl. Table 1 [Major 1]: could the authors also add the outcome itself (average phase shift per time)? I would consider it important to see this outcome in combination with the stimulation settings that are already mentioned in the table.

And finally, two very minor comments/typos:

c) Lines 668-669: "Because an electric field is considered quasistatic, it can be divided into spatial and temporal components" reads like E-fields in general can be considered quasistatic, which is obviously not true. I'd suggest writing something like "Because the electric fields in our study are varying at low frequencies <100 Hz, they can be considered quasistatic [...]"

d) Supplementary Fig. S6 does not explicitly mention that it refers to experiment 1/human data.

Reviewer #2 (Remarks to the Author):

I would like to thank the authors for their thoughtful responses. I believe that the manuscript has undergone significant improvements, and the additional analyses have substantially strengthened the findings.

I have just a couple of extra comments that should be addressed.

1 - Regarding experiment 2, my concern was not really about the size (or SNR) of the spikes but rather their temporal stability (I imagine the value of the SNR per se is mainly determined by the user's choice of the lambda parameter). If the amplitude of a spike decreases over time, it may increase the likelihood of either not detecting spikes from that specific putative neuron or picking up spikes from other putative neurons nearby. Progressively picking up spikes from nearby neurons could lead to apparent phase shifts. The ISI distributions depicted in supfig14, particularly the examples in the bottom row, seem somewhat exponential, which is not expected for well-isolated neurons (although the temporal scale should be rescaled to see clear refractory periods). One way to address this concern is to plot the spike amplitude (or SNR) as a function of time (for instance using the same windows used to estimate the phase). Another approach would involve computing an average of spikes in temporal windows and then perform the cross-correlation between the initial average waveform and the other ones. Ideally these approaches should show that the phase changes that are reported in the manuscript do not track fluctuations in the stability of the waveforms (either amplitude of the spikes or modifications of the cross-correlations in time).

2 - Regarding my previous comment 12, I apologize for the error in referencing. I intended to refer to Figure 3B instead of 2B. I was interested in knowing whether the distribution of phase precessions, when considering all units and not just those defined as having significant phase precession, was different from what would be expected by chance.

The new analysis presented, which demonstrates the absence of precession in the non-stimulation periods, should effectively address my comment. In essence, it shows that without stimulation, there is no "spontaneous" precession. Is that correct? How does the full distribution of phase precessions across neurons (without any threshold over entrained units) look like in stimulated vs unstimulated periods?

We would like to, once again, thank both reviewers for their efforts. Our response to the raised questions can be found below. Changes in the manuscript and supplementary materials are marked in red font.

Reviewer #1 (Remarks to the Author):

The authors provided additional descriptions, simulations and analyses that relieve my major concerns, and I agree with the authors that they have significantly improved their manuscript. I have two minor/clarification points left [related to previous major points]:

Reply: Thank you for reviewing our manuscript for a second time and we appreciate the additional feedback. In summary, we performed additional analyses to demonstrate phase resets in the NHP data and the absence of phase shifts in humans in the absence of tACS.

a) 'Gradual' shifts in spike timing / phase precession.

The authors argue with different strategies [Major 2a] for human data (shuffling of phase information) and animal data (comparison to sham condition). Could the authors explain this choice, or should one assume that the respective other strategy would not have been successful?

Reply: The two approaches we used depend on the experimental design. In the human experiment 'phase' is an experimentally manipulated factor. We used our closed-loop system to specifically target four discrete phases (peak, fall, trough, and rise). However, in the NHP experiment 'phase' is an observational outcome factor. As such, using the approach of reshuffling phase conditions in the NHP data is not possible since there are no discrete phase conditions.

Alternatively, for the human data phase changes can be investigated in our previous published data (Wischnewski et al., 2022), which we re-analyzed here. Similar to the current study, in our previous data we also applied 150 TMS pulses across 4 blocks, however, without tACS. We analyzed potential phase preferences and shifts with respect to a virtual tACS signal at the alpha and beta frequency. Specifically, normalized MEP amplitude and phase with regards to the virtual tACS signal was determined. Subsequently the averaged polar vector was calculated per window in the same way as was done in the original data (Figure 2). The results demonstrate no phase preference, nor an apparent shift of MEPs with regards to the virtual tACS signal. We added this data to the supplementary materials (Supplementary Fig. S6) and refer to the results on page 6, lines 128-130:

“Also, we investigated potential phase shifts to a virtual tACS signal in a dataset where no active stimulation was applied, and found no apparent phase preference, nor phase shifts (Supplementary Fig. S6).”

Supplementary Fig. S6. Polar plots of changes in phase of primary motor cortex excitability with regards to a virtual tACS signal, without active stimulation. For this control analysis data was taken from Wischnewski et al. (2022). This dataset follows the same structure as the current experiment, namely 4 blocks of 150 TMS pulses (600 in total), with each block lasting approximately 6 minutes. However, no actual tACS was applied. We applied an alpha and beta virtual tACS signal and extracted the corresponding phases at which the TMS pulse occurred. Subsequently polar vectors were calculated based on phase and MEP amplitude averaged over data per time window, blocks and participants. Each dot represents the polar vector of 55 trials, which equates to approximately two minutes. The sliding window moves in steps of 5 trials (~12 seconds), resulting in 20 windows. On the left the results are shown for the alpha (9.92 Hz) virtual tACS in gray. On the right the results are shown for the beta (20.24 Hz) virtual tACS in gray. Further, the main results with active tACS, as presented in Figure 2 of the main article, are shown transparently for reference. In contrast to active tACS, for the virtual tACS no apparent phase preference, nor phase shift was observed.

Together, the permutation method presented previously (in the same participants) and the present results (in a different sample) suggest that the tACS-induced phase shifts are unlikely to be random.

Similarly, the 'resets' [Major 2c] of phase shifts were discussed for the human data, but not for the animal data. Were the phase shifts similarly consistent across stimulation blocks in the animal data?

Reply: This is a great point. Therefore, we inspected the phase preference in the non-human primate data. We investigated the phase at the end of a block (window 20, denoted as W_{k20} , where the W stands for window and k for the block number) to the phase at the start of the next block (window 1 of the subsequent block, denoted as W_{k+11}). We found

that the phase change between blocks is of similar magnitude, but opposite direction than the tACS-related phase shift of each block (window 20 – window 1). This suggests that on average across neurons a reset of phase occurred between stimulation blocks. The data is presented in Supplementary Fig. 24, and we refer to this data on page 10, lines 199-201:

“Furthermore, phase shifts were consistent across the four blocks and phase was reset between blocks (Supplementary Fig. S24), comparable to what was observed in the human data (Supplementary Fig. S7).”

Supplementary Figure 24. In purple: Phase change during alpha and beta tACS per block for all clockwise and counter-clockwise neurons. For this the preferred phase at the first (1) window (W) of block k was subtracted from the preferred phase of the last (20) window (W) of block k . Naturally, for neurons that show an overall clockwise phase shift (main text, Figure 3), per block neurons also mostly display a clockwise shift (phase change < 0). Similarly, for neurons that shows an overall counter-clockwise phase shift (main text, Figure 3), per block neurons also mostly display a counter-clockwise shift (phase change > 0). In blue: Phase change between stimulation blocks (at rest). For this the preferred phase of the final (20) window (W) of a block (k) was subtracted from the first (1) window (W) of the subsequent block ($k+1$). On average phase change between

blocks is of similar magnitude but in opposed direction compared to the change during blocks (purple). This suggests a phase reset between blocks, as was also observed in humans (Supplementary Fig. 7).

b) Suppl. Table 1 [Major 1]: could the authors also add the outcome itself (average phase shift per time)? I would consider it important to see this outcome in combination with the stimulation settings that are already mentioned in the table.

Reply: Thank you for the suggestion. We added a row on the main finding per experiment. Please see below and **Supplementary Table 1**.

	Experiment 1 (human)	Experiment 2 (NHP)	Experiment 3a (population modelling)	Experiment 3b (microcircuit modelling)
tACS intensity	1 mA	1 mA	n/A	n/A
Max. e-field	0.31 mV/mm	0.84 mV/mm	1.5 mV/mm	3 mV/mm
tACS duration	4 x 6 min (24 min total)	4 x 6 min (24 min total)	6 min	5 min
tACS frequency	On average: 9.92 Hz and 20.24 Hz	10 Hz and 20 Hz	10 Hz and 20 Hz	10 Hz and 20 Hz
tACS montage	2 PiStim (Ø 3.14 cm ²) electrodes 7 cm anterior and posterior of M1	2 PiStim (Ø 3.14 cm ²) electrodes Fp1 and PO3	2 PiStim (Ø 3.14 cm ²) electrodes 7 cm anterior and posterior of M1 (modelled)	2 PiStim (Ø 3.14 cm ²) electrodes 7 cm anterior and posterior of M1 (modelled)
Outcome measure	MEP	Single-unit activity	Single-unit activity	Single-unit activity
Main finding	70.9° (alpha tACS) and 57.4° (beta tACS) phase shift on averaged normalized MEP over 6 min	Phase shifts in 15/46 (alpha tACS) and 13/48 neurons (beta tACS), ranging between ~15° and ~80° over 6 min	AC-induced increase in phase locking in anterior and posterior motor cortex depending on electric field direction	~60° (alpha tACS) and ~80° (beta tACS) phase shift on M1 spiking due to increased NMDA weights between BA6 and BA4

And finally, two very minor comments/typos:

c) Lines 668-669: “Because an electric field is considered quasistatic, it can be divided into spatial and temporal components” reads like E-fields in general can be considered quasistatic, which is obviously not true. I’d suggest writing something like “Because the electric fields in our study are varying at low frequencies <100 Hz, they can be considered quasistatic [...]”.

Reply: Thank you, we changed the sentence as suggested (page 24, lines 669-671): “Because the electric fields in our study are varying at low frequencies <100 Hz, they can be considered quasistatic and can be divided into spatial and temporal components.”

d) Supplementary Fig. S6 does not explicitly mention that it refers to experiment 1/human data.

Reply: We now added this information: “Phase shifts of MEPs per stimulation block in experiment 1.”

Reviewer #2 (Remarks to the Author):

I would like to thank the authors for their thoughtful responses. I believe that the manuscript has undergone significant improvements, and the additional analyses have substantially strengthened the findings.

I have just a couple of extra comments that should be addressed.

Reply: We thank the reviewer for the assessing our manuscript for a second time and we fully appreciate the additional comments. In summary, we changed the supplementary materials to include the change of spiking over time and included an analysis on phase shifts in all entrained neurons.

1 - Regarding experiment 2, my concern was not really about the size (or SNR) of the spikes but rather their temporal stability (I imagine the value of the SNR per se is mainly determined by the user's choice of the lambda parameter). If the amplitude of a spike decreases over time, it may increase the likelihood of either not detecting spikes from that specific putative neuron or picking up spikes from other putative neurons nearby. Progressively picking up spikes from nearby neurons could lead to apparent phase shifts. The ISI distributions depicted in supfig14, particularly the examples in the bottom row, seem somewhat exponential, which is not expected for well-isolated neurons (although the temporal scale should be rescaled to see clear refractory periods). One way to address this concern is to plot the spike amplitude (or SNR) as a function of time (for instance using the same windows used to estimate the phase). Another approach would involve computing an average of spikes in temporal windows and then perform the cross-correlation between the initial average waveform and the other ones. Ideally these approaches should show that the phase changes that are reported in the manuscript do not track fluctuations in the stability of the waveforms (either amplitude of the spikes or modifications of the cross-correlations in time).

Reply: We fully agree with the reviewer that a large change in the spike amplitude over time will definitely affect the results (either picking up artifacts or nearby neurons). Accordingly, we modified the Supplementary Figure S15 (before it was S14) by 1) adding at the bottom right the evolution of the spike amplitude over the 20 time windows and 2) rescaling the temporal scale on the ISI figure for a clearer visibility. We can observe that the spike amplitude remains relatively stable over time.

Supplementary Fig. S15. Examples of cluster of single units recorded. A) Extracellular spike waveforms of 4 isolated units are shown with additional features. The ISI of the cell is shown in the bottom right with the total number of spikes detected during the recording period. On the left of the ISI is the change in the spike amplitude over the 20 time windows. In the top left corner is the amplitude of the spike (mean \pm standard deviation, unit: μV). The thick black line represents the mean average waveform. The black line in the bottom left of each graph represents 200 μs . B) Histogram of the signal-to-noise ratio (SNR) of the cells recorded (3.50 ± 0.73 , mean \pm standard deviation).

2 - Regarding my previous comment 12, I apologize for the error in referencing. I intended to refer to Figure 3B instead of 2B. I was interested in knowing whether the distribution of phase

precessions, when considering all units and not just those defined as having significant phase precession, was different from what would be expected by chance.

The new analysis presented, which demonstrates the absence of precession in the non-stimulation periods, should effectively address my comment. In essence, it shows that without stimulation, there is no "spontaneous" precession. Is that correct? How does the full distribution of phase precessions across neurons (without any threshold over entrained units) look like in stimulated vs unstimulated periods?

Reply: In order to investigate the absence of phase precession during the pre-stimulation period, we computed the average of the window-to-window phase shift difference for all entrained neurons (Rayleigh test, $p < 0.05$) for the three conditions. The results show a broader range of phase shifts during alpha and beta stimulation compared to the no stimulation condition. This suggests that, across all entrained neurons, tACS generally induces larger phase shifts compared to spontaneous phase shifts. We present these results in Supplementary Fig. 23 and refer to these results on page 10, lines 199-200:

"Analysis of phase shifts across all entrained neurons confirms this observation (Supplementary Fig. 23)."

Supplementary Fig. S23. Distribution of the average window-to-window phase shift difference for all entrained neurons for alpha tACS (blue, A), beta tACS (orange, B), and no stimulation (gray, A and B). Distributions of phase shifts when no stimulation is applied are quite narrow suggesting quite limited spontaneous phase changes. Alternatively, during tACS distributions are broader, suggesting a wider range of phase shifts induced by tACS.

REVIEWERS' COMMENTS

Reviewer #1 (Remarks to the Author):

All of my points have been fully addressed, and I would like to congratulate the authors on their amazing work.

Reviewer #2 (Remarks to the Author):

I would like to thank the authors for the further clarifications. I believe the manuscript largely improved throughout the reviewing process and I therefore have no further comments.